# NOISY SCRUBBER: UNLEARNING USING NOISY REPRESENTATIONS

## ABSTRACT

Machine Unlearning (MU) aims to remove the influence of specific data points from trained models, with applications ranging from privacy enforcement to debiasing and mitigating data poisoning. Although exact unlearning ensures complete data removal via retraining, this process is computationally intensive, motivating the development of efficient approximate unlearning methods. Existing approaches typically modify model parameters, which limits scalability, introduces instability, and requires extensive tuning. We propose Noisy Scrubber, a novel MU framework that learns to inject perturbations into the latent representations rather than modifying model parameters. To show Noisy Scrubber attains approximate unlearning we theoretically establish bounds on the parameter gap between original and exact unlearned model, as well as on the output discrepancy between Noisy Scrubber and exact unlearning. Empirical results on CIFAR-10, CIFAR-100, and AGNews demonstrate that Noisy Scrubber closely matches exact unlearning while being significantly more efficient, reducing unlearning gaps to 0.024, 0.129, and 0.006, respectively. Moreover, membership inference evaluations confirm that Noisy Scrubber removes information comparably to retraining. Our approach scales across model families in both vision and text, and introduces a flexible, attachable noise module that enables on-demand and reversible unlearning.

## 1 INTRODUCTION

Deep learning models are increasingly deployed in real-world applications, achieving notable success driven by advances in computational resources, data availability, and neural network architectures. Training these models typically demands substantial compute and large-scale datasets, which may accidentally include copyrighted content, private user data, biases, or other sensitive information. Recent regulations, such as the General Data Protection Regulation (GDPR) and the California Consumer Privacy Act (CCPA), grant users the right to request deletion of their data from machine learning systems Nguyen et al. (2022); Wang et al. (2024). However, removing specific data points from trained deep neural networks presents significant challenges. The highly non-convex nature of neural networks makes it difficult to trace the influence of specific data points on model parameters. Furthermore, learned representations are densely encoded in the model weights, where a single neuron captures multiple unrelated concept phenomena known as polysemanticity Elhage et al. (2022). This superposition complicates the removal of targeted information, as adjusting one neuron may disrupt model utility. These challenges highlight the need for developing machine learning pipelines that support strong performance while enabling efficient and reliable data deletion.

*Machine Unlearning* (MU) emerged as an area that aims to remove the influence of specific data points from the trained model. One direct approach is *exact unlearning*, which involves retraining the model on the data to be retained. While retraining the model from scratch provides strong guarantees, it is computationally expensive. Therefore, recent research has focused on developing efficient and fast *approximation unlearning* algorithms Fan et al. (2023); Jia et al. (2023); Izzo et al. (2021); Thudi et al. (2022a); Neel et al. (2021). Although approximate unlearning methods provide computational benefit, they do not provide provable guarantees of unlearning effectiveness, resulting performance gap compared to exact unlearning Golatkar et al. (2020). While effectiveness cannot be proven formally, it can be evaluated empirically through metrics such as accuracy on forget, retain, and test sets and membership inference attack Shokri et al. (2017); Liu et al. (2022) without requiring

data, model, or algorithmic assumptions as in certified unlearning Guo et al. (2019). Therefore, developing an approximate unlearning algorithm requires balancing the inherent trade-offs between forgetting quality, model utility, and efficiency.

Despite the advantage over exact unlearning, prior approximation unlearning methods exhibit significant performance variance because of hyperparameter selection, stochasticity from model parameter optimisation and evaluation metric selection Fan et al. (2023); Jia et al. (2023). These approaches require direct modification of the trained model parameters, which limits scalability and demands extensive hyperparameter tuning that can introduce instability, particularly in large models. In contrast, *We present Noisy Scrubber, a MU framework that avoids direct modification of trained model parameters. Instead, it employs a neural network to generate noise that is injected into the latent representation space, aligning the model's predictions with the exact unlearned model's predictions.* We show Noisy Scrubber attains approximate unlearning by theoretically establishing bounds on the parameter gap between original and exact unlearned model, as well as on the output discrepancy between Noisy Scrubber and exact unlearning.

We empirically demonstrate the effectiveness of Noisy Scrubber in classification tasks for both classwise and random data forgetting across text and vision modalities. Our results show that Noisy Scrubber exhibits strong unlearning capabilities and outperforms several state-of-the-art unlearning methods. Experiments conducted on models of varying scales highlight its effectiveness and efficiency, particularly for larger models. Membership inference attacks further confirm that Noisy Scrubber achieves information removal comparable to exact unlearning, thereby reducing the risk of leaked forget information. Additionally, we show that Noisy Scrubber remains effective even with limited subset of retain and forget samples. We summarise our contributions as follows:

- We introduce Noisy Scrubber, a novel MU framework that achieves unlearning by injecting perturbations into latent representations, avoiding direct parameter modification.
- We derive bounds on (i) the parameter difference between trained and retrained models, and (ii) the output discrepancy between Noisy Scrubber and exact unlearning to show Noisy Scrubber attains approximate unlearning.
- The noise module is lightweight, attachable, and independently training, enabling on-demand and reversible unlearning.
- We demonstrate that Noisy Scrubber effectively approximates exact unlearning across CIFAR-10, CIFAR-100, and AGNews, ranging from small CNNs to large architectures such as ResNet-101, SWIN Transformer, and BERT. It achieves negligible membership inference leakage, and outperforms state-of-the-art MU baselines under limited retain data.

## 2 MACHINE UNLEARNING AND EVALUATION

### 2.1 UNLEARNING PROBLEM FORMULATION

MU is the process of algorithmically removing the contribution of specific data points from a pretrained machine learning model Cao & Yang (2015); Bourtoule et al. (2021); Nguyen et al. (2022); Wang et al. (2024). The objective is to produce an updated model that behaves as if it had never been trained on that data. It is critical for applications such as enforcing privacy regulations, like the right to be forgotten, and removing outdated or incorrect information from the model Nguyen et al. (2022); Wang et al. (2024). Formally, let $\theta^0$ denote the initial model parameters. We define the model as a composite function $f : \mathcal{X} \times \mathbb{R}^\theta \to \mathbb{R}^k$, $f(x, \theta) = g(h(x, \theta_{\text{feat}}), \theta_{\text{cls}})$, where $h$ is the feature extractor with parameters $\theta_{\text{feat}}$, $g$ is the classification head with parameters $\theta_{\text{cls}}$, and the full parameter set is $\theta = (\theta_{\text{feat}}, \theta_{\text{cls}})$. A model is trained on a dataset $D$ using a learning algorithm $\mathcal{A}$ to yield the trained parameters $\theta^t = \mathcal{A}(D, \theta^0)$. We partition the training set $D$ into two disjoint subsets: the *forget set* $F \subseteq D$, containing the data to be unlearned, and the *retain set* $R = D \setminus F$. The primary goal of a MU algorithm is to find an unlearned model parameters, $\theta^u$, using the trained model parameters $\theta^t$, the forget set $F$, and the retain set $R$, such that $f(., \theta^u)$ is a close approximation of the retrained model $f(., \theta^r)$ where $\theta^r = \mathcal{A}(R, \theta^0)$. In the context of classification, the choice of the forget set $F$ defines various unlearning scenarios Bourtoule et al. (2021); Graves et al. (2021). Specifically, *classwise forgetting* refers to the removal of all training data associated with a particular class, while *random forgetting* involves unlearning a randomly selected subset of training samples, which may contain samples across all classes.

## 2.2 EXACT AND APPROXIMATE UNLEARNING

Unlearning algorithms are generally classified into two types: exact and approximate unlearning. Exact unlearning algorithms produce a model parameters $\theta^u$ that is identical to the one obtained by retraining $\theta^r$. For neural networks, this usually means retraining from scratch or retraining a subset of models affected by the data removal in the ensemble. However, exact methods entail a large computational overhead, especially when training deep neural networks. For an unlearning algorithm to be practical, the unlearning process must be significantly more computationally efficient than retraining the model from scratch on the retain set. Therefore, approximate unlearning algorithms are preferred in practice, as they are a proxy for full retraining, hence more computationally efficient. These methods use the trained model parameters $\theta^t$, the retain set $R$, and the forget set $F$ to produce an updated model parameters that behaves similarly to a retrained model parameters $\theta^r$. However, the gain in computational efficiency comes at the cost of reduced effectiveness in MU.

## 2.3 EVALUATING APPROXIMATE UNLEARNING

Unlearning evaluation can be divided into two categories: (i) assessing the extent of unlearning and model utility, and (ii) evaluating the indistinguishability of the model and its outputs. An effective unlearning algorithm must balance the trade-off among forgetting quality, model utility, and unlearning efficiency. The research community uses several empirical measures to capture these trade-offs. Zhao et al. (2024) proposed the "tug-of-war" (ToW) metric, which captures relative accuracy differences between unlearned and retrained models across forget, retain, and test sets, thereby quantifying the trade-off between forgetting quality and model utility. Privacy is typically evaluated using Membership Inference Attacks (MIA), where an adversary attempts to distinguish forget samples from unseen data. The MIA score is defined as the proportion of forget samples identified as test data by the adversary and MIA-GAP quantifies the deviation of this score from the retrained baseline. Computational efficiency is reflected in runtime efficiency (RTE), defined as the relative speedup of an unlearning algorithm over full retraining. Finally, indistinguishability is evaluated both at the parameter level, using the $\ell_2$ norm of parameter differences, and at the output level, via the Jensen–Shannon divergence (JSD) between predictions of unlearned and retrained models. Together, these metrics provide a holistic view of effectiveness, privacy, scalability, and similarity to exact unlearning. Further details of all evaluation metrics is provided in Appendix A.3.

## 3 RELATED WORK

In recent years, there has been significant interest in MU. The goal of MU is to modify trained models in order to remove the influence of particular data points, originally motivated to prevent privacy breaches Ginart et al. (2019); Neel et al. (2021); Ullah et al. (2021); Sekhari et al. (2021). While retraining the model from scratch (exact unlearning) provides strong guarantees, it is generally computationally infeasible for large datasets and models. To address this challenge, recent research has focused on developing efficient approximate unlearning algorithms. These methods seek to remove the effects of targeted data without incurring the high computational costs of full retraining. Among these, the foundational methods FT and GA were discussed in Appendix A.2. However, GA has been observed to significantly degrade overall model utility. Building on these, NegGrad+ Kurmanji et al. (2023) integrates both FT and GA in a joint optimisation framework: it minimises the loss on the retain set $R$ while maximising the loss on the forget set $F$. SCRUB Kurmanji et al. (2023), an extension by the same authors, frames the problem as knowledge distillation Hinton et al. (2015). Here, a student network is trained to imitate the teacher's behaviour on $R$ but disobey the teacher on $F$. $\ell_1$-sparse unlearning Jia et al. (2023) introduces an $\ell_1$ penalty during fine-tuning, inspired by model pruning, to promote sparsity in parameter updates. Other strategies leverage model sensitivity. Fisher Forgetting (FF) Becker & Liebig (2022); Golatkar et al. (2020) adds Gaussian noise to model parameters, where the covariance is determined by the Fisher information matrix. However, calculating the Fisher information at scale is computationally intensive and offers limited parallelizability on modern hardware. Influence Unlearning (IU) Izzo et al. (2021); Koh & Liang (2017) uses influence functions to estimate how individual points affect model parameters, connecting this work to $\epsilon$-$\delta$ Guo et al. (2019) forgetting, but often relies on strong model and training assumptions Guo et al. (2019). Additionally, re-labelling-based methods erase knowledge by reassigning labels to the forget set $F$ using samples from a prior distribution, for example, a uniform distribution used in the

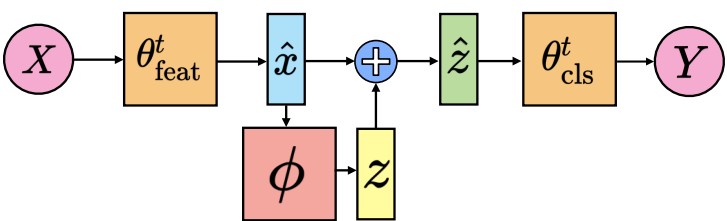

Figure 1: Noisy Scrubber Framework

random-label method. Saliency Unlearning (SalUn) Fan et al. (2023) selectively updates only the most relevant, or salient parameters, partitioning model weights into those affected by $F$ and those that are not. In contrast to prior works that require fine-tuning or updating model parameters, our approach does not modify model parameters during unlearning, thereby avoiding the high computational cost and instability associated with updating the main model parameters. Instead, we train a noise module that perturbs the model's learned representations, selectively corrupting the representations of forget samples while leaving the retain samples unaffected. To show Noisy Scrubber attains approximate unlearning, we derived bounds on parameter difference between trained and retrained models, and output discrepancy between Noisy Scrubber and exact unlearning.

## 4 NOISY SCRUBBER: ERASE WITH NOISY CORRUPTION

### 4.1 PROBLEM DEFINITION

We focus on the problem of approximate unlearning in the classification setting, considering both classwise and random forgetting scenarios. Let the forget set be denoted as $F \subset D$, and the retain set as $R \subset D \setminus F$. We consider a practical setting where, instead of using the entire set $D \setminus F$ as the retain set, we choose a representative subset $R$. This assumption is particularly relevant when complete access to the training dataset is unavailable, and only a limited number of samples from the training distribution are available. Our goal is to derive model parameters $\theta^u = \mathcal{U}(F, R, \theta^t)$ using the unlearning algorithm $\mathcal{U}$ that closely approximates the ideal retrained model. For evaluation, this retrained model parameters $\theta^r$ is defined as parameters obtained by training from scratch on the complete set $D \setminus F$.

**Proposition 1** (Parameter Difference Bound). *Let $D = R \cup F$ be the full dataset, partitioned into a retain set $R$ and a forget set $F$, where $R \cap F = \emptyset$. Let the empirical risk functions be defined as $L_D(\theta) = \frac{1}{|D|} \sum_{x \in D} l(x, \theta)$ and $L_R(\theta) = \frac{1}{|R|} \sum_{x \in R} l(x, \theta)$. The corresponding optimal parameters are defined as $\theta^t = \arg\min_\theta L_D(\theta)$ and $\theta^r = \arg\min_\theta L_R(\theta)$, both obtained by performing $T$ steps of gradient descent from the same initialisation $\theta_0$ with step size $\alpha$.*

*Assume the loss function $l(x, \theta)$ satisfies the following properties:*

1. *It is continuously differentiable in $\theta$ for every $x$.*

2. *It is $M$-Lipschitz smooth with respect to $\theta$.*

3. *The per-sample gradient is bounded, i.e., $|\nabla l(x, \theta)| \leq G$ for some constant $G > 0$.*

*Then, the Euclidean distance between the optimal parameters is upper bounded as:*

$$\|\theta^t - \theta^r\| \leq \frac{2G|F|}{M|D|}(e^{\alpha MT-1} - 1)$$

*Under assumption of $\mu$-strong convexity of loss function $l(x, \theta)$ upper bound reduces to:*

$$\|\theta^t - \theta^r\| \leq \frac{G|F|}{\mu|D|}$$

Proposition 1 establishes that the magnitude of the parameter perturbation induced by removing a subset $F$ scales linearly with the subset size $|F|$ and decays at a rate inversely proportional to the total dataset size $|D|$. This highlights that model parameters are more stable in well-conditioned settings with large datasets, however remain sensitive to the removal of small subsets containing high-influence samples. Further, imposing a strong convexity assumption leads to a significantly tighter bound, indicating that strong convexity functions as an effective regularizer. In practice, strong convexity can be introduced during training through the addition of L2 regularisation. The complete proof of Proposition 1 is provided in the Appendix A.4.

## 4.2 UNLEARNING WITH NOISY SCRUBBER

**Noisy Representations.** We begin with motivating the rationale behind noisy representations for MU. Prior work on data uncertainty has primarily focused on the noisy labels, noisy inputs, and out-of-distribution (OOD) samples Li et al. (2021; 2022); Geng et al. (2021); Li et al. (2023). Motivated by data uncertainty, we consider the central question: *Can we perturb the representations of forget samples in a way that makes them highly uncertain, without affecting the representations of retain samples?* Targeting input representations is necessary, as the MU setting does not permit prior assumptions about the presence of input or output noise. Given that the exact unlearned model parameters $\theta^r$ produces highly uncertain outputs for forget samples, while maintaining retain set performance, our objective is formulated as perturbing input representations so that they align with the outputs of $\theta^r$.

**Noisy Scrubber formulation.** Let $\theta^t$ be decomposed into two components: feature extractor parameters $\theta^t_{\text{feat}}$ and classifier parameters $\theta^t_{\text{cls}}$. $\theta^t_{\text{feat}}$ generates a representation vector $\hat{x} = h(x, \theta^t_{\text{feat}}) \in \mathbb{R}^d$ for a given input $x$, and $\theta^t_{\text{cls}}$ maps the representation $\hat{x}$ to a prediction over $k$ classes, i.e., $y = g(\hat{x}, \theta^t_{\text{cls}})$. Noisy Scrubber aims to inject noise vector $\epsilon$ into latent representations of $\theta^t$, i.e., $h(x, \theta^t_{\text{feat}}) + \epsilon$ such that its outputs align with outputs of $\theta^r$. Mathematically our objective is defined as finding a noise vector $\epsilon$ such that:

$$g(h(x, \theta^t_{\text{feat}}) + \epsilon, \theta^t_{\text{cls}}) \approx g(h(x, \theta^r_{\text{feat}}), \theta^r_{\text{cls}})$$

We can find $\epsilon$ by solving the following optimisation problem:

$$\epsilon = \arg\min_{\hat{\epsilon} \in \mathbb{R}^d} \|g(h(x, \theta^t_{\text{feat}}) + \hat{\epsilon}, \theta^t_{\text{cls}}) - g(h(x, \theta^r_{\text{feat}}), \theta^r_{\text{cls}})\| \tag{1}$$

Deriving a closed-form solution for $\epsilon$ requires strong assumptions on $g$, such as invertibility, which is generally intractable in the case of neural networks. Instead, we can use a gradient descent-based algorithm to obtain the optimal value of $\epsilon$.

**Proposition 2** (Output Difference Bound). *Consider a full dataset $D$ partitioned into a retain set $R$ and a forget set $F$ such that $D = R \cup F$ and $R \cap F = \emptyset$. We define two parameter vectors, $\theta^t$ and $\theta^r$, which are the outputs of training on the empirical risks $L_D(\theta)$ and $L_R(\theta)$, respectively. Both are obtained by running $T$ iterations of gradient descent with a learning rate $\alpha$ from a common initialisation $\theta_0$.*

*Assume the following conditions hold:*

1. *The model $f(x, \theta)$ is $L$-Lipschitz continuous with respect to its parameter $\theta$ for all $x \in \mathcal{X}$.*

2. *The per-sample loss function $l(x, \theta)$ is $M$-smooth with respect to $\theta$.*

3. *The per-sample gradient is bounded, i.e., $|\nabla l(x, \theta)| \leq G$ for some constant $G > 0$.*

*Then, for any input $x \in \mathcal{X}$ and $\epsilon = \arg\min_{\hat{\epsilon} \in \mathbb{R}^d} \|g(h(x, \theta^t_{feat}) + \hat{\epsilon}, \theta^t_{cls}) - g(h(x, \theta^r_{feat}), \theta^r_{cls})\|$, the difference in the model's output is upper bounded by:*

$$\|g(h(x, \theta^t_{feat}) + \epsilon, \theta^t_{cls}) - f(x, \theta^r)\| \leq \frac{2GL|F|}{M|D|}(e^{\alpha MT} - 1)$$

Proposition 2 shows that the perturbed model output is closer to the exact unlearned model compared to the unperturbed case demonstrating approximate unlearning. The bound provides a formal guarantee that output discrepancy can be controlled and reduced through the optimal perturbation $\epsilon$. The proof of Proposition 2 is provided in the Appendix A.5.

**Noisy Scrubber.** In order to perform unlearning we require to calculate optimal $\epsilon$ using Equation (1). Directly solving Equation (1) for each sample requires access to the parameters of $\theta^r$ and involves iteratively optimising per sample, which is computationally expensive. To address this, we train a neural network $p(.,\phi)$ to approximate the solution of this optimisation problem. The network $p(.,\phi)$, takes the intermediate representation $\hat{x} = h(x, \theta^t_{feat})$, as input and generates a noise vector $z = p(\hat{x}, \phi)$. This noise vector is added to the model's intermediate representation $\hat{z} = \hat{x} + z$, and the resulting perturbed representation is subsequently processed by the downstream layers for prediction $y_{\text{noise}} = g(\hat{z}, \theta^t_{\text{cls}})$. The combination of the original model parameters $\theta^t$ and the noise module parameters $\phi$ constitutes the unlearned model parameters $\theta^u = [\theta^t, \phi]$. Figure 1 shows the proposed MU pipeline. This noise module is *independent* of the model and can be *easily attached or detached*, which also allows us to forget learned knowledge for a specific period of time. Furthermore, this approach improves the efficiency of unlearning since only the noise module requires training, which is significantly more efficient than updating all parameters of model.

Before presenting the training details of the noise module parameters $\phi$, we introduce *Uniform-Except-One* (**UEO**) distribution $\mathcal{U}^k_c$ over $k$ classes, where a specified class $c \in \{1, 2, .., k\}$ are assigned equal probability mass. For each class index $i \in \{1, \ldots, k\}$, the probability mass function is given by:

$$\mathcal{U}^k_c(i) = \begin{cases} 0 & \text{if } i = c, \\ \frac{1}{k-1} & \text{otherwise.} \end{cases}$$

In the classification setting, we avoid relying on the model parameters $\theta^r$ to optimise Equation (1), by assuming that an exact unlearned model output a UEO distribution for forget set samples reflecting maximal uncertainty, while preserving the same output distribution as $\theta^t$ for retain samples. This assumption captures the fact that the exact unlearned model has no exposure to the forgotten data, and therefore assigning zero probability to the forgotten class or a uniform distribution over all classes corresponds to the maximum-entropy distribution. Therefore, we train the noise module with a KL-divergence based contrastive loss (KLC) (Equation 2), which encourages $p(.,\phi)$ to generate noisy representations $\hat{z}$ such that $g(., \theta^t_{\text{cls}})$ preserves the original output distribution on retain samples while aligning forget sample outputs with the UEO distribution.

$$\text{KLC} = (1 - \lambda)\text{KL}(y_{\text{noisy}}, y_{\text{clean}}) + \lambda\text{KL}\left(y_{\text{noisy}}, \mathcal{U}^k_{\arg\max\{y_{\text{clean}}\}}\right) \tag{2}$$

where, $\lambda = 0$ if $x \in R$ and $\lambda = 1$ if $x \in F$, $y_{\text{noisy}}$ is prediction by $\theta^t$ after noisy perturbation in representation, $y_{\text{clean}}$ is prediction by $\theta^t$ without perturbation and $\mathcal{U}^k_c$ represents UEO distribution where, $k$ denotes the total number of classes in the classification task. During training, model parameters $\theta^t$ is frozen, intermediate representations of input $x$ are passed to the noise module $p(.,\phi)$, which generates noisy representation $\hat{z}$. During the backward pass, only the noise module is updated such that the KLC loss Equation (2) is minimised.

**Distill Noisy Scrubber (Distill-NS).** Unlike the retrained parameters $\theta^r$, Noisy-Scrubber augments the model with an additional noise module $\phi$, yielding parameters $\theta^u = [\theta^t, \phi]$. This distinct parameterization makes the two models distinguishable which makes evaluation of proximity between $\theta^u$ and $\theta^r$ difficult, and increases susceptibility to inference attacks Yang et al. (2024). To address this problem, we employ knowledge distillation Hinton et al. (2015) to obtain a single, unified model $\theta^u$ that does not include the additional parameters from the noise module. Knowledge distillation is a technique for transferring knowledge from a larger teacher model to a smaller student model. In our approach, the unlearned parameters $\theta^u = \theta_{\text{teacher}} = [\{\theta^t_{\text{feat}}, \theta^t_{\text{cls}}\}, \phi]$ serves as the teacher. A separate copy of the original trained model parameters $\theta^t$ acts as the student model $\theta_{\text{student}} = \{\theta^t_{\text{feat}}, \theta^t_{\text{cls}}\}$. The student model is trained to mimic the teacher using a knowledge distillation loss (KD) defined as:

$$\text{KD} = \alpha\,\text{KL}(f(.\theta_{\text{student}}), f(., \theta_{\text{teacher}})) + \beta\,\text{MSE}(o_{\text{student}}, o_{\text{teacher}}) + \gamma\,\text{CE}(\arg\max\{f(., \theta_{\text{student}})\}, y) \tag{3}$$

where $o_{\text{student}} = h(., [\theta^t_{\text{feat}}]_{\text{student}})$ denotes the student's intermediate representation, $o_{\text{teacher}} = h(., [\phi(\theta^t_{\text{feat}})]_{\text{teacher}})$ is the noisy representation produced by the teacher, $y$ represents the true class label, CE denotes cross-entropy loss and MSE denotes mean square error. The student model is trained to minimise KD-loss using both the forget set $F$ and the retain set $R$. This yields a single unlearned model parameters $\theta^u$ which excludes noise parameters $\phi$.

Table 1: Performance comparison of MU methods in the class-wise forgetting scenario. Results are reported as $a \pm b$, denoting the mean $a$ and standard deviation $b$ over 5 independent trials.

| Methods | Metrics | CIFAR10 | CIFAR100 | AGNews |
|---|---|---|---|---|
| Retrain | **RTE (sec)** | 137.27 | 234.61 | 3064.68 |
| FT | **ToW** | 0.983±0.00 | 0.308±0.00 | 0.748±0.02 |
| | **MIA-Gap** | 0.002±0.00 | 0.556±0.00 | 0.006±0.09 |
| | **RTE (sec)** | 8.757±0.15 | 46.912±2.46 | 199.404±19.87 |
| GA | **ToW** | 0.434±0.00 | 0.554±0.00 | 0.204±0.00 |
| | **MIA-Gap** | 0.000±0.00 | 0.002±0.00 | 0.000±0.00 |
| | **RTE (sec)** | 5.620±1.80 | 38.831±0.61 | 143.615±0.21 |
| $\ell_1$-Sparse | **ToW** | 0.976±0.00 | 0.316±0.00 | 0.871±0.01 |
| | **MIA-Gap** | 0.000±0.00 | 0.546±0.00 | 0.035±0.00 |
| | **RTE (sec)** | 9.352±1.26 | 48.989±1.88 | 201.630±11.21 |
| IU | **ToW** | 0.075±0.00 | 0.021±0.0 | 0.481±0.00 |
| | **MIA-Gap** | 0.000±0.00 | 0.000±0.00 | 0.209±0.00 |
| | **RTE (sec)** | 8.980±0.06 | 46.188±1.89 | 77.856±0.91 |
| NegGrad+ | **ToW** | 0.894±0.00 | 0.756±0.00 | 0.955±0.00 |
| | **MIA-Gap** | 0.000±0.00 | 0.002±0.00 | 0.000±0.00 |
| | **RTE (sec)** | 8.162±0.31 | 26.373±0.15 | 136.951±1.59 |
| Random-Label | **ToW** | 0.921±0.00 | 0.346±0.00 | 0.738±0.13 |
| | **MIA-Gap** | 0.000±0.00 | 0.433±0.00 | 0.001±0.00 |
| | **RTE (sec)** | 11.035±0.39 | 49.457±1.172 | 244.402±3.35 |
| SCRUB | **ToW** | 0.786±0.00 | 0.469±0.0 | 0.902±0.00 |
| | **MIA-Gap** | 0.002±0.00 | 0.332±0.0 | 0.000±0.00 |
| | **RTE (sec)** | 9.400±0.53 | 49.069±0.58 | 241.720±3.18 |
| SALUN | **ToW** | 0.918±0.00 | 0.352±0.00 | 0.628±0.18 |
| | **MIA-Gap** | 0.000±0.00 | 0.433±0.00 | 0.000±0.00 |
| | **RTE (sec)** | 14.891±5.49 | 71.998±42.65 | 336.329±5.52 |
| Noisy-Scrubber | **ToW** | 0.976±0.00 | 0.871±0.00 | 0.994±0.00 |
| | **MIA-Gap** | 0.000±0.00 | 0.000±0.00 | 0.000±0.00 |
| | **RTE (sec)** | 15.658±0.13 | 24.710±0.45 | 95.002±1.19 |
| Distill-NS | **ToW** | 0.968±0.00 | 0.564±0.00 | 0.993±0.00 |
| | **MIA-Gap** | 0.000±0.00 | 0.186±0.00 | 0.000±0.00 |
| | **RTE (sec)** | 22.317±2.70 | 27.901±3.19 | 269.674±0.17 |

# 5 EXPERIMENT AND ANALYSIS

## 5.1 EXPERIMENT SETUPS

**Dataset and Models:** We evaluate unlearning using standard classification benchmark datasets from both image and text domains, including MNIST LeCun et al. (2010), CIFAR10 and CI-FAR100 Krizhevsky & Hinton (2009), AGNews Zhang et al. (2015b), and DBPedia Zhang et al. (2015a). For the MNIST and CIFAR10 datasets, we utilize simple CNN LeCun et al. (1995)architectures, while for CIFAR100, we employ ResNet He et al. (2016) and SWIN transformer models Liu et al. (2021). For AGNews and DBPedia, we utilized BERT Devlin et al. (2019) model.
**Evaluation Metrics:** We evaluate unlearning performance using the ToW metric (Equation 4), MIA-GAP, and RTE. We implement MIA using prediction confidence attack based method Yeom et al. (2018); Song & Mittal (2021). To evaluate indistinguishability, we used JSD (Equation 6) and $\ell_2$ norm of model parameters. Further details of all evaluation metrics is provided in Appendix A.3.
**Unlearning Setup:** We investigate two unlearning scenarios: class-wise forgetting and random data forgetting. For our experiments, the forget set $F$ contains 1000 samples, while the retain set $R$ comprises 2000 samples. For class-wise forgetting, we randomly select 1000 samples from a single class to form the forget set $F$, and 2000 samples randomly drawn from the remaining classes to create the retain set $R$. For random forgetting, we select 1000 samples at random from the dataset to form $F$, and 2000 samples from the remaining data to form $R$. We compare our approach against baselines and the state-of-art, including FT, GA, NegGrad+, $\ell$1-sparse, Random-label, SCRUB, and SALUN, implementing each methods by following their official repositories. Hyperparameters for these methods are tuned for each dataset, with the corresponding values detailed in the Appendix A.6.3. All experiments were conducted on *NVIDIA RTX 4070 (12GB)* GPU.

Table 2: Indistinguishability of different MU methods in the class-wise forgetting scenario. Results are reported as $a \pm b$, with $a$ representing the mean and $b$ the standard deviation computed over 5 independent trials.

| Methods | Metrics | CIFAR10 | CIFAR100 | AGNews |
|---------|---------|---------|----------|--------|
| FT | **JSD** | $0.3000 \pm 0.00$ | $0.793 \pm 0.00$ | $0.3350 \pm 0.02$ |
| | $\ell_2$-**Score** | $55.125 \pm 0.00$ | $125.00 \pm 0.00$ | $18.182 \pm 0.05$ |
| GA | **JSD** | $1.716 \pm 0.02$ | $2.05 \pm 0.00$ | $3.550 \pm 0.02$ |
| | $\ell_2$-**Score** | $54.67 \pm 0.00$ | $124.99 \pm 0.00$ | $12.99 \pm 0.00$ |
| $\ell_1$-Sparse | **JSD** | $0.249 \pm 0.00$ | $0.777 \pm 0.00$ | $0.263 \pm 0.00$ |
| | $\ell_2$-**Score** | $53.617 \pm 0.06$ | $124.539 \pm 0.00$ | $124.923 \pm 0.06$ |
| IU | **JSD** | $16.498 \pm 0.00$ | $20.40 \pm 0.00$ | $0.467 \pm 0.00$ |
| | $\ell_2$-**Score** | $150.37 \pm 0.00$ | $134.955 \pm 0.00$ | $11.225 \pm 0.00$ |
| NegGrad+ | **JSD** | $0.388 \pm 0.00$ | $1.576 \pm 0.00$ | $0.451 \pm 0.00$ |
| | $\ell_2$-**Score** | $54.705 \pm 0.00$ | $125.09 \pm 0.00$ | $14.478 \pm 0.02$ |
| Random-Label | **JSD** | $0.416 \pm 0.00$ | $0.762 \pm 0.00$ | $0.351 \pm 0.00$ |
| | $\ell_2$-**Score** | $56.534 \pm 0.00$ | $125.132 \pm 0.00$ | $19.532 \pm 0.14$ |
| SCRUB | **JSD** | $0.371 \pm 0.00$ | $0.804 \pm 0.00$ | $0.137 \pm 0.00$ |
| | $\ell_2$-**Score** | $55.228 \pm 0.00$ | $125.060 \pm 0.00$ | $14.721 \pm 0.09$ |
| SALUN | **JSD** | $0.404 \pm 0.00$ | $0.756 \pm 0.00$ | $0.323 \pm 0.00$ |
| | $\ell_2$-**Score** | $56.260 \pm 0.00$ | $125.069 \pm 0.00$ | $18.835 \pm 0.31$ |
| **Distill-NS** | **JSD** | $0.248 \pm 0.00$ | $0.882 \pm 0.00$ | $0.079 \pm 0.00$ |
| | $\ell_2$-**Score** | $56.092 \pm 0.00$ | $124.946 \pm 0.00$ | $11.256 \pm 0.00$ |
| **Noisy Scrubber** | **JSD** | $\mathbf{0.938 \pm 0.00}$ | $1.988 \pm 0.00$ | $0.411 \pm 0.00$ |

## 5.2 EXPERIMENTAL RESULTS

**Noisy Scrubber scrubs away information from trained model.** Table 1 presents a comprehensive evaluation of our proposed method *Noisy Scrubber* and its distilled variant, *Distill-NS*. For reference, the ideal ToW and MIA-Gap values are 1 and 0 for exact unlearning. Any approximate unlearning algorithm that exhibit minimum deviation from these ideal values is considered superior. On CIFAR10, CIFAR100, and AGNews datasets, *Noisy Scrubber* achieves ToW gaps of 0.066, 0.129, and 0.024 relative to retraining, demonstrating its strong ability to approximate the performance of exact unlearning, outperforming all baselines and prior state-of-the-art unlearning techniques. In terms of privacy leakage, measured by MIA-Gap, Noisy Scrubber obtains values very close to zero in all three datasets, signifying negligible vulnerability to MIA and forget information leakage. Additional results on datasets, models, and results on random forgetting scenarios are provided in Appendix Sections A.8 and A.7. The *Distill-NS* yields a single unified unlearned model, enabling the computation of a pseudo-distinguishability score for Noisy Scrubber. In terms of distinguishability, shown in Table 2 Distill-NS achieves highly competitive results on CIFAR10, CIFAR100, and AGNews, while demonstrating effective unlearning. In terms of computational efficiency, both *Noisy Scrubber* and *Distill-NS* achieve notable runtime improvements over conventional retraining and surpass prior unlearning approaches in balancing speed and efficacy. Significant reductions in runtime are achieved for large architectures, such as ResNet101 and Swin Transformer on CIFAR100 and BERT on AGNews. While improvements are less in CIFAR10 due to its smaller model size, *Noisy Scrubber* still maintains competitive runtime efficiency. Although IU achieves the fastest runtime on AGNews, its significantly lower unlearning effectiveness and privacy scores undermine its practical utility. Overall, these result establish *Noisy Scrubber* as a scalable, privacy-preserving, and effective machine unlearning solution across a range of modalities and model architectures.

**Comparison with number of retain samples.** We evaluate baseline and prior state-of-the-art unlearning methods alongside Noisy Scrubber across varying retain set sizes, ranging from 10 to 2000 samples. This evaluation aims to assess the effectiveness of each method when limited amount of retain data is available. Figure 2 demonstrates that increasing the number of retain samples on CIFAR10 leads to improved performance for Noisy-Scrubber, which converges at 500 samples and outperforms both baselines and prior state-of-the-art methods. On CIFAR100, performance initially rises with more retain samples but exhibits a sharp decline at 1000 samples. This is because the forget set contains only 500 samples (representing a single class) and with 1000 retain samples, the noise module becomes biased towards the retain set, resulting in higher forget set accuracy and drop

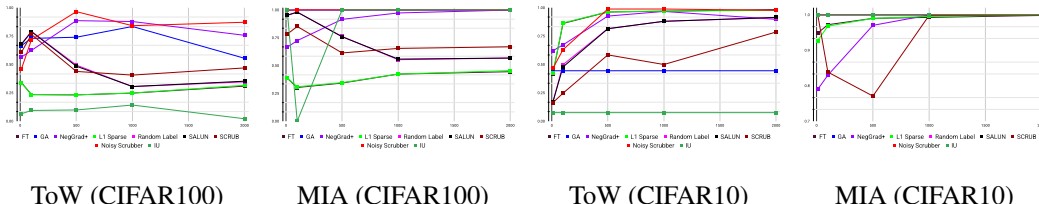

ToW (CIFAR100)     MIA (CIFAR100)     ToW (CIFAR10)     MIA (CIFAR10)

Figure 2: Comparison of the impact of number of retain samples on ToW and MIA performance for CIFAR10 and CIFAR100 datasets

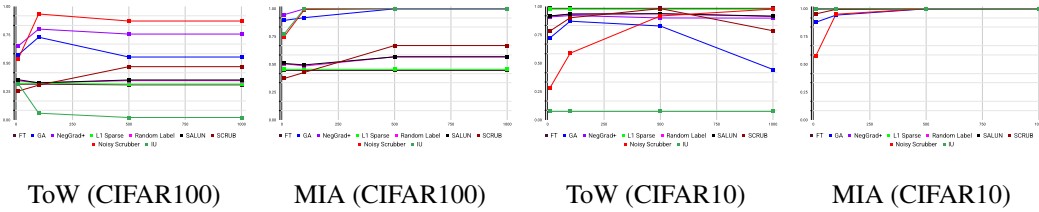

ToW (CIFAR100)     MIA (CIFAR100)     ToW (CIFAR10)     MIA (CIFAR10)

Figure 3: Comparison of the impact of number of forget samples on ToW and MIA performance for CIFAR10 and CIFAR100 datasets.

in ToW. As the number of retain samples increases further, Noisy-Scrubber's performance recovers, contrasting with GA, NegGrad+, and IU, whose performance continues to deteriorate. It is also noted that, for both CIFAR10 and CIFAR100, Noisy-Scrubber starts with lower ToW compared to other methods when trained with very few retain samples, indicating that the noise module initially requires few data samples for effective learning. Figure 2 also compare the MIA scores as the number of retain samples increases. Noisy Scrubber consistently maintains MIA scores close to 1 on both CIFAR10 and CIFAR100, demonstrating its strong resilience against membership inference attacks and forget information leakage.

**Comparison with number of forget samples.** This evaluation aims to assess the effectiveness of baseline and prior state-of-the-art unlearning methods when limited amount of forget data is available. Figure 3 illustrates that increasing the number of forget samples leads to higher ToW scores on both CIFAR10 and CIFAR100 datasets. For small number of forget samples, ToW values are lower compared to other methods in both datasets, as the noise module requires a sufficient quantity of forget samples for effective training. Beyond a certain threshold in number of forget samples, the ToW metric saturates. Similarly, Figure 3 shows that the MIA scores of Noisy Scrubber improve with an increasing number of forget samples. This improvement occurs because the noise module becomes more proficient at corrupting forget sample representations, making them indistinguishable from test data to membership inference attackers.

## 6 CONCLUSION

This work introduced *Noisy Scrubber*, a novel framework for machine unlearning that injects targeted noise into latent representations rather than modifying model parameters. To show Noisy Scrubber attains approximate unlearning we theoretically establish bounds on the parameter gap between original and exact unlearned model, as well as on the output discrepancy between Noisy Scrubber and exact unlearning. To bypass costly per-sample optimization for computing optimal perturbations, we introduce a noise module trained with a KL divergence-based contrastive loss to approximate these perturbations. Extensive experiments on CIFAR-10, CIFAR-100, and AGNews, ranging from small CNNs to large architectures such as ResNet-101, SWIN Transformer, and BERT demonstrate that Noisy Scrubber consistently approximates exact unlearning while outperforming state-of-the-art baselines in accuracy retention, privacy protection, and runtime efficiency. These findings highlight representation-level perturbations as a promising direction for scalable unlearning. Future research may explore extending Noisy Scrubber to continual unlearning, scenarios with extremely limited forget/retain data, and large-scale foundation models.

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

# A APPENDIX

## A.1 BROADER IMPACT

Machine Unlearning (MU) research holds significant promise for enabling users to request deletion of their data from trained models, enhancing model safety by removing inappropriate or obsolete information. In this work, we develop a scalable approximate unlearning method that achieves strong data forgetting, maintains high utility, and preserves generalisation. This is particularly relevant as model and dataset sizes continue to grow, and in practical scenarios where models may be iteratively retrained by third parties, often without access to the original training data. Additionally, updating all parameters in a large model is unstable and requires a lot of tuning. We tried to address all these issues in this work.

## A.2 BASELINE APPROXIMATION UNLEARNING

We next review two baseline approximation unlearning techniques.

♣ **Fine-tuning (FT)**: FT Golatkar et al. (2020); Warnecke et al. (2021) fine-tunes the original model $\theta^t$ on the retain set $R$ for small number of training epochs to yield $\theta^u$. It leverages the concept of catastrophic forgetting from continual learning Parisi et al. (2019), where training on $R$ initiates the model $\theta^t$ to forget information related to $F$.

♣ **Gradient Ascent (GA)**: GA Graves et al. (2021); Thudi et al. (2022a) fine-tunes the original model $\theta^t$ on the forget set $F$ with the objective of maximizing the loss for samples in $F$ to yield $\theta^u$.

## A.3 UNLEARNING EVALUATION

Prior studies have examined MU performance Golatkar et al. (2020) Thudi et al. (2022a) Graves et al. (2021), however, using a single evaluation metric may not fully reflect the performance of MU Thudi et al. (2022b). Based on our review of prior evaluation methods, we focus on the following evaluation methods and metrics used in our experimental investigation.

♣ **Accuracy-based metric:** In classification tasks, model utility is typically measured by accuracy on the retain and test sets, where higher accuracy denotes better utility. Forgetting quality is empirically evaluated by the accuracy on the forget set, with lower accuracy implying better unlearning. Ideally, post-unlearning accuracy on the forget and retain sets should closely match that of a model retrained from scratch. Zhao et al. (2024) introduced the *"tug-of-war"* (**ToW**) metric (Equation 4), which computes the relative accuracy differences between unlearned and retrained models on forget, retain, and test sets.

$$\text{ToW}(\theta^u, \theta^r, F, R, D_{test}) = (1 - \text{da}(\theta^u, \theta^r, F)) \cdot (1 - \text{da}(\theta^u, \theta^r, R)) \cdot (1 - \text{da}(\theta^u, \theta^r, D_{\text{test}})) \quad (4)$$

where $a(\theta, D)$ denotes accuracy on $D$ of model $\theta$ and $\text{da}(\theta^u, \theta^r, D) = |a(\theta^u, D) - a(\theta^r, D)|$ is absolute difference between accuracy of model $\theta^u$ and $\theta^r$ on $D$. ToW favours models whose

accuracy closely matches that of retrained models across forget, retain and test sets. ToW values range from 0 to 1, where higher scores indicate better unlearning.

♣ **Membership Inference Attack Score (MIA Score):** In privacy literature, MIA Shokri et al. (2017) assesses whether a given sample is part of the training dataset or not by examining model outputs (e.g. confidence, loss values). MIA can be utilised to assess forgetting quality. To calculate MIA we define an attacker that attempts to distinguish between samples in the forget set from the samples in the never-seen distribution (test set not part of the training process). The failure of the attacker in distinguishing never-seen samples from the forget set acts as a measure of forgetting quality. MIA is calculated as $\frac{TN_F}{|F|}$ where, $TN_F$ denote number of true negative predicted by MIA model on forget set and $|F|$ denote number of samples in forget set. MIA score ranges from 0 to 1, where a higher value signifies that more forget set samples are predicted as non-training data, reflecting better unlearning. Ideally, the MIA score of the unlearned model should closely match that of a model retrained from scratch. *MIA-GAP* is defined to compare MIA performance between the unlearned model and the retrained model, which is the absolute difference between MIA scores of the unlearned model and the retrained model. Zhao et al. (2024) proposed the *ToW-MIA* metric (Equation 5), which measures forget quality using MIA instead of accuracy.

$$\text{TOW-MIA}(\theta^u, \theta^r, F, R, D_{test}) = (1 - \text{dm}(\theta^u, \theta^r, F)) \cdot (1 - \text{da}(\theta^u, \theta^r, R)) \cdot (1 - \text{da}(\theta^u, \theta^r, D_{\text{test}})) \quad (5)$$

where $m(\theta, D)$ denotes MIA on $D$ of model $\theta$ and the first term $dm(\theta^u, \theta^r, F) = |m(\theta^u, F) - m(\theta^r, F)|$ represents the absolute MIA performance difference between unlearned and retrained models. Unlike ToW, which measures forgetting quality via accuracy, ToW-MIA assesses it using the MIA score. Similar to ToW, ToW-MIA also ranges from 0 and 1, where a higher score indicates better unlearning.

♣ **Run-time efficiency (RTE):** It evaluates the runtime efficiency of the unlearning algorithm ($\mathcal{U}$) by comparing it to the retraining baseline. Specifically, the time required by the retrained model is used as a reference, and the speedup achieved by the approximate unlearning algorithm is measured relative to this baseline.

♣ **Indistinguishability measures:** The indistinguishability measures similarity between unlearned model $\theta^u$ and retrained model $\theta^r$. One standard approach is to compute the $\ell_2$ norm of the difference between $\theta^u$ and $\theta^r$, given by $\|\theta^u - \theta^r\|$. A smaller $\ell_2$ norm indicates higher similarity between the two models. Additionally, the indistinguishability of the model's outputs can be quantified using the *Jensen-Shannon divergence* (**JSD**) (Equation 6) between the prediction on data samples $D$ by model $\theta^u$ and $\theta^r$.

$$\text{JSD}(\theta^u, \theta^r, D) = \frac{1}{2}\big(\text{KL}(\theta^u(D)\|Q) + \text{KL}(\theta^r(D)\|Q))\big) \quad (6)$$

where $Q = \frac{1}{2}(\theta^u(D) + \theta^r(D))$ and KL denotes the Kullback-Leibler divergence. Lower JS divergence values indicate that the outputs of the unlearned and retrained models are more difficult to distinguish.

### A.4 PROOF OF PROPOSITION 1

Let $D = R \cup F$ be a dataset, where $R \cap F = \emptyset$. Let the empirical risk functions be defined as $L_D(\theta) = \frac{1}{|D|}\sum_{x \in D} l(x, \theta)$ and $L_R(\theta) = \frac{1}{|R|}\sum_{x \in R} l(x, \theta)$. The corresponding optimal parameters are defined as $\theta^t = \arg\min_\theta L_D(\theta)$ and $\theta^r = \arg\min_\theta L_R(\theta)$, both obtained by performing $T$ steps of gradient descent from the same initialisation $\theta_0$ with step size $\alpha$.

Assume the loss function $l(x, \theta)$ satisfies the following properties:

1. It is continuously differentiable in $\theta$ for every $x$.

2. It is $M$-Lipschitz smooth with respect to $\theta$.

3. The gradient norm is bounded for any single sample, i.e., $\|\nabla l(x, \theta)\| \leq G$ for some constant $G > 0$.

If both parameters $\theta^t, \theta^r$ are updated using GD, let us denote the updates as follows:

$$\theta^t_{i+1} = \theta^t_i - \alpha \nabla L_D(\theta^t_i)$$
$$\theta^r_{i+1} = \theta^r_i - \alpha \nabla L_R(\theta^r_i)$$

Let $\delta_i = \theta^t_i - \theta^r_i$, then find $\|\delta_T\| = \|\theta^t_T - \theta^r_T\|$

let $\theta^t$ converges at $T_1$ ie.., $\theta^t = \theta^t_{T_1}$, $\theta^r$ converges at $T_2$ ie.., $\theta^r = \theta^r_{T_2}$
we assume $T$ rounds such that $T > T_1$ & $T > T_2$,
which still implies: $\theta^t = \theta^t_T$ and $\theta^r = \theta^r_T$ as $\nabla L_D(\theta) = 0$ and $\nabla L_R(\theta) = 0$, $\forall$ steps $> T_1$ & $T_2$

*Proof.* We start with

$$\delta_{i+1} = \theta^t_{i+1} - \theta^r_{i+1}$$
$$= \theta^t_i - \alpha \nabla L_D(\theta^t_i) - \theta^r_i + \alpha \nabla L_R(\theta^r_i)$$
$$= (\theta^t_i - \theta^r_i) - \alpha(\nabla L_D(\theta^t_i) - \nabla L_R(\theta^r_i))$$
$$\|\delta_{i+1}\| = \|(\theta^t_i - \theta^r_i) - \alpha(\nabla L_D(\theta^t_i) - \nabla L_R(\theta^r_i))\|$$
$$\leq \|\theta^t_i - \theta^r_i\| + \alpha\|\nabla L_D(\theta^t_i) - \nabla L_R(\theta^r_i))\|$$

After applying the triangular inequality:

$$\|\delta_{i+1}\| \leq \|\delta_i\| + \alpha\|\nabla L_D(\theta^t_i) - \nabla L_R(\theta^r_i)\| \tag{7}$$

Find $\|\nabla L_D(\theta^t_i) - \nabla L_D(\theta^r_i)\| \leq ?$

Add & Subtract $\nabla L_R(\theta^t_i)$

$$\|\nabla L_D(\theta^t_i) - \nabla L_D(\theta^r_i)\| \leq \|\nabla L_D(\theta^t_i) - \nabla L_R(\theta^t_i) + \nabla L_R(\theta^t_i) - \nabla L_R(\theta^r_i)\|$$
$$\leq |\nabla L_D(\theta^t_i) - \nabla L_R(\theta^t_i)\| + \|\nabla L_R(\theta^t_i) - \nabla L_R(\theta^r_i)\|$$

Now, we have two terms to be bounded
Term A : $\|\nabla L_D(\theta^t_i) - \nabla L_R(\theta^t_i)\|$ and
Term B : $\|\nabla L_R(\theta^t_i) - \nabla L_R(\theta^r_i)\|$
**Bounding Term A:**

$$\|\nabla L_D(\theta^t_i) - \nabla L_R(\theta^t_i)\| = \|\tfrac{1}{|D|}\sum_{x\in D}\nabla l(x,\theta^t_i) - \tfrac{1}{|R|}\sum_{x\in R}\nabla l(x,\theta^t_i)\|$$
$$= \|\tfrac{1}{|D|}\sum_{x\in F}\nabla l(x,\theta^t_i) + \tfrac{1}{|D|}\sum_{x\in R}\nabla l(x,\theta^t_i) - \tfrac{1}{|R|}\sum_{x\in R}\nabla l(x,\theta^t_i)\|$$
$$= \|\tfrac{1}{|D|}\sum_{x\in F}\nabla l(x,\theta^t_i)\| + \|(\tfrac{1}{|D|} - \tfrac{1}{|R|})(\sum_{x\in R}\nabla l(x,\theta^t_i))\|$$
$$= \|\tfrac{1}{|D|}\sum_{x\in F}\nabla l(x,\theta^t_i)\| + \|(\tfrac{|R|-|D|}{|D||R|})\sum_{x\in R}\nabla l(x,\theta^t_i)\|$$

Upon applying triangular inequality, we get

$$\|\nabla L_D(\theta^t_i) - \nabla L_R(\theta^t_i)\| \leq \|\tfrac{1}{|D|}\sum_{x\in F}\nabla l(x,\theta^t_i)\| + \tfrac{|F|}{|D||R|}\sum_{x\in R}\|\nabla l(x,\theta^t_i)\|$$
$$\leq \tfrac{1}{|D|}\sum_{x\in F}G + \tfrac{|F|}{|D||R|}\sum_{x\in R}G$$
$$= \tfrac{|F|G}{|D|} + \tfrac{|F||R|}{|D||R|}G$$
$$= \tfrac{2|F|}{|D|}G$$

**Bounding Term B:**

$$\|\nabla L_R(\theta^t_i) - \nabla L_R(\theta^r_i)\| = \|\tfrac{1}{|R|}\sum_{x\in R}\nabla l(x,\theta^t_i) - \tfrac{1}{|R|}\sum_{x\in R}\nabla l(x,\theta^r_i)\|$$
$$\leq \tfrac{1}{|R|}\sum_{x\in R}\|\nabla l(x,\theta^t_i) - \nabla l(x,\theta^r_i)\|$$

Using the definition of M-smooth, we get

$$\|\nabla L_R(\theta_i^t) - \nabla L_R(\theta_i^r)\| \leq \frac{1}{|R|} \sum_{x \in R} M\|\theta_i^t - \theta_i^r\|$$

$$= \frac{M}{|R|}|R|\|\delta_t\|$$

$$= M\|\delta_t\|$$

Therefore, upon substituting both the bounds, we get the final bound as follows

$$\|\nabla L_D(\theta_i^t) - \nabla L_D(\theta_i^r)\| \leq \frac{2|F|G}{D} + M\|\delta_i\|$$

Therefore, we substitute the bound in eq. 7, to get

$$\|\delta_{i+1}\| \leq \|\delta_i\| + \alpha M\|\delta_i\| + \frac{\alpha 2|F|G}{|D|}$$

$$\leq (1 + \alpha M)\|\delta_i\| + \frac{\alpha 2 G|F|}{|D|}$$

when $i + 1 = T$,

$$\|\delta_T\| \leq (1 + \alpha M)\|\delta_{T-1}\| + \frac{\alpha 2 G|F|}{|D|}$$

Similarly,

$$\|\delta_{T-1}\| \leq (1 + \alpha M)\|\delta_{T-2}\| + \frac{2\alpha G|F|}{|D|}$$

On substituting $\|\delta_{T-1}\|$ in $\|\delta_T\|$, we get

$$\|\delta_T\| \leq (1 + \alpha M)[(1 + \alpha M)\|\delta_{T-2}\| + \frac{2\alpha G|F|}{|D|}] + \frac{2\alpha G|F|}{|D|}$$

$$\leq (1 + \alpha M)^2\|\delta_{T-2}\| + \frac{2\alpha G|F|}{|D|}](1 + \alpha M) + \frac{2\alpha G|F|}{|D|}$$

Similarly, we substitute $\|\delta_{T-2}\|$

$$\|\delta_T\| \leq (1 + \alpha M)^2[(1 + \alpha M)\|\delta_{T-3}\| + \frac{2\alpha G|F|}{|D|}] + \frac{2\alpha G|F|}{|D|}(1 + \alpha M) + \frac{2\alpha G|F|}{|D|}$$

$$\leq (1 + \alpha M)^3\|\delta_{T-3}\| + (1 + \alpha M)^2\frac{2\alpha G|F|}{|D|} + (1 + \alpha M)\frac{2\alpha G|F|}{|D|} + \frac{2\alpha G|F|}{|D|}$$

$$\leq (1 + \alpha M)^3\|\delta_{T-3}\| + \frac{2\alpha G|F|}{|D|}[1 + (1 + \alpha M) + (1 + \alpha M)^2]$$

Unrolling for $T$ steps yields:

$$\|\delta_T\| \leq (1 + \alpha m)^T\|\delta_0\| + \frac{2\alpha G|F|}{|D|}[1 + (1 + \alpha m) + (1 + \alpha m)^2 + \cdots + (1 + \alpha m)^{T-1}]$$

Sum of geometric progression:

$$S_n = \frac{a(r^n - 1)}{r - 1}, a = 1, r = (1 + \alpha m)$$

So the sum inside the bound becomes:

$$\frac{(1 + \alpha m)^T - 1}{\alpha m}$$

Therefore,

$$\|\delta_T\| \leq (1 + \alpha m)^T\|\delta_0\| + \frac{2\alpha|F|G}{|D|}[\frac{(1+\alpha m)^T - 1}{\alpha M}]$$

$$\leq (1 + \alpha m)^T\|\delta_0\| + \frac{2|F|G}{m|D|}[(1 + \alpha m)^T - 1]$$

Assuming $\|\delta_0\| = \|\theta_0^t - \theta_0^r\| = 0$, i.e. $\theta_0^t = \theta_0^r$,

$$\|\theta_T^t - \theta_T^r\| = \|\theta^t - \theta^r\| \leq \frac{2|F|G}{m|D|}\left[(1 + \alpha m)^T - 1\right]$$

Since for $x > 0$,

$$(1 + x) \leq 1 + x + \frac{x^2}{2!} + \frac{x^3}{3!} + \cdots = e^x$$
$$\leq e^x$$

As, $\alpha \geq 0$, $m \geq 0 \Rightarrow \alpha M \geq 0$ Therefore,

$$1 + \alpha M \leq e^{\alpha M}$$
$$(1 + \alpha m)^T \leq e^{\alpha m T}$$

Final bound:

$$\|\theta^t - \theta^r\| \leq \frac{2|F|G}{m|D|}(e^{\alpha m T} - 1)$$

Now assuming loss function $l(x, \theta)$ is $\mu$-strongly convex. Since the empirical risk $L_D(\theta)$ is an average of $\mu$-strongly convex functions, it is also $\mu$-strongly convex. For any two points $\theta_1, \theta_2$, the following inequality holds for $\mu$-strongly convex function:

$$\|\theta_1 - \theta_2\| \leq \frac{1}{\mu}\|\nabla L_D(\theta_1) - \nabla L_D(\theta_2)\|$$

Setting $\theta_1 = \theta^t$ and $\theta_2 = \theta^r$, and using the first-order optimality condition $\nabla L_D(\theta) = 0$, we get:

$$\|\theta^t - \theta^r\| \leq \frac{1}{\mu}\|0 - \nabla L_D(\theta^r)\| = \frac{1}{\mu}\|\nabla L_D(\theta^r)\| \tag{8}$$

$\|\nabla L_D(\theta^r)\| \leq ?$

We expand this term using the dataset definition $D = R \cup F$:

$$\nabla L_D(\theta^r) = \frac{1}{|D|}\left(\sum_{x \in R} \nabla l(x, \theta^r) + \sum_{x \in F} \nabla l(x, \theta^r)\right)$$

Since $\theta^r$ is the minimizer of $L_R(\theta)$, we know that $\sum_{x \in R} \nabla l(x, \theta^r) = 0$. This simplifies the expression to:

$$\nabla L_D(\theta^r) = \frac{1}{|D|}\sum_{x \in F} \nabla l(x, \theta^r)$$

Taking the norm, applying the triangle inequality, and using the gradient bound assumption:

$$\|\nabla L_D(\theta^r)\| \leq \frac{1}{|D|}\sum_{x \in F}\|\nabla l(x, \theta^r)\| \leq \frac{1}{|D|}\sum_{x \in F} G = \frac{|F|G}{|D|}$$

Finally, substituting this result back into Equation equation 8, we arrive at our final bound:

$$\|\theta^t - \theta^r\| \leq \frac{1}{\mu}\left(\frac{|F|G}{|D|}\right) = \frac{|F|G}{\mu|D|}$$

$\square$

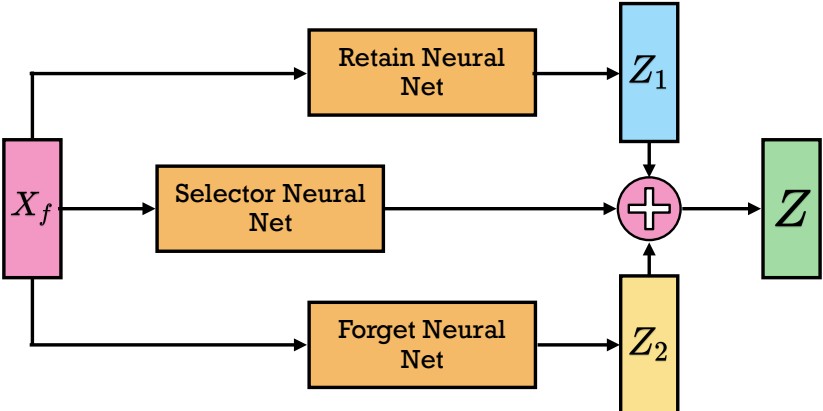

Figure 4: Noise Model

## A.5    PROOF OF PROPOSITION 2

Consider a full dataset $D$ partitioned into a retain set $R$ and a forget set $F$ such that $D = R \cup F$ and $R \cap F = \emptyset$. We define two parameter vectors, $\theta^t$ and $\theta^r$, which are the outputs of training on the empirical risks $L_D(\theta)$ and $L_R(\theta)$, respectively. Both are obtained by running $T$ iterations of gradient descent with a learning rate $\alpha$ from a common initialisation $\theta_0$.

Assume the following conditions hold:

1. The model $f(x, \theta)$ is $L$-Lipschitz continuous with respect to its parameter $\theta$ for all $x \in \mathcal{X}$.

2. The per-sample loss function $l(x, \theta)$ is $M$-smooth with respect to $\theta$.

3. The per-sample gradient is uniformly bounded, i.e., $\|\nabla l(x, \theta)\| \leq G$.

*Proof.*

$$\min_{\hat{\epsilon} \in \mathbb{R}^d} \|g(h(x, \theta^t_{\text{feat}}) + \hat{\epsilon}, \theta^t_{\text{cls}}) - g(h(x, \theta^r_{\text{feat}}), \theta^r_{\text{cls}})\| = \|g(h(x, \theta^t_{\text{feat}}) + \epsilon, \theta^t_{\text{cls}}) - g(h(x, \theta^r_{\text{feat}}), \theta^r_{\text{cls}})\|$$

for any $\hat{\epsilon} \in \mathbb{R}^d$ we have,

$$\|g(h(x, \theta^t_{\text{feat}}) + \epsilon, \theta^t_{\text{cls}}) - g(h(x, \theta^r_{\text{feat}}), \theta^r_{\text{cls}})\| \leq \|g(h(x, \theta^t_{\text{feat}}) + \hat{\epsilon}, \theta^t_{\text{cls}}) - g(h(x, \theta^r_{\text{feat}}), \theta^r_{\text{cls}})\|$$

$\hat{\epsilon} = 0$ also satisfies the above inequality

$$\|g(h(x, \theta^t_{\text{feat}}) + \epsilon, \theta^t_{\text{cls}}) - g(h(x, \theta^r_{\text{feat}}), \theta^r_{\text{cls}})\| \leq \|g(h(x, \theta^t_{\text{feat}}), \theta^t_{\text{cls}}) - g(h(x, \theta^r_{\text{feat}}), \theta^r_{\text{cls}})\|$$
$$\|g(h(x, \theta^t_{\text{feat}}) + \epsilon, \theta^t_{\text{cls}}) - g(h(x, \theta^r_{\text{feat}}), \theta^r_{\text{cls}})\| \leq \|f(x, \theta^t) - f(x, \theta^r)\|$$

since, $f$ is $L$-Lipschitz with respect to parameters, we have $\|f(x, \theta^t) - f(x, \theta^r)\| \leq L\|\theta^t - \theta^r\|$ Applying Proposition 1, this directly yields the stated bound. $\square$

## A.6    IMPLEMENTATION DETAILS

### A.6.1    NOISE MODEL ARCHITECTURE

We implemented the noise model as depicted in Figure 4. For all experiments, the noise model architecture was fixed as follows: both the retain and forget neural networks are linear layers with a hidden dimension of 512, while the selector network is a linear layer with a hidden dimension of 64. The selector outputs a probability used to determine the weighted combination of the retain ($z_1$) and forget ($z_2$) network outputs, yielding the final noise value. The following is a PyTorch implementation snippet for the model:

```
class NoisyLatchModel(nn.Module):
```

```
def __init__(self, input_dim, hidden_dim, attn_hidden_dim, *args,
             **kwargs):
    super().__init__(*args, **kwargs)
    self.un_model_forget = nn.Sequential(
        nn.Linear(input_dim, hidden_dim),
        nn.ReLU(),
        nn.Linear(hidden_dim, input_dim),
    )
    self.un_model_retain = nn.Sequential(
        nn.Linear(input_dim, hidden_dim),
        nn.ReLU(),
        nn.Linear(hidden_dim, input_dim),
    )
    self.attn_wts = nn.Sequential(
        nn.Linear(input_dim, attn_hidden_dim),
        nn.ReLU(),
        nn.Linear(attn_hidden_dim, 2),
    )

def forward(self, repr):
    attn = torch.softmax(self.attn_wts(repr), dim=1)
    retain = self.un_model_retain(repr)
    forget = self.un_model_forget(repr)
    z = torch.einsum("bn,bni->bi", attn,
                     torch.stack((retain, forget), dim=1))
    repr_ = repr + z
    return z, repr_, attn
```

### A.6.2   TRAINING ORIGINAL MODELS

We trained four model architectures across five datasets: simple CNNs for MNIST and CIFAR10, ResNet101 and SWIN Transformers for CIFAR100, and BERT for AGNews and DBPedia. Models were trained with early stopping based on validation accuracy. For CNNs, we used the AdamW optimiser with a weight decay of $1 \times 10^{-5}$ while for ResNet101, SWIN and BERT, a weight decay of $1 \times 10^{-2}$ was applied. A cosine annealing warmup restart (CSWR) learning rate scheduler was used to tune the learning rate during training. The initial learning rate was set to $1 \times 10^{-3}$ for CNN, ResNet101, and SWIN, and $2 \times 10^{-5}$ for BERT.

### A.6.3   TRAINING DETAILS OF MACHINE UNLEARNING METHODS

We compared Noisy Scrubber with eight approximate unlearning methods as well as an exact un-learning baseline based on retraining. Hyperparameters for each method were carefully selected for each dataset and architecture. In contrast, Noisy Scrubber's architecture and hyperparameters were fixed across all experiments using values identified via tuning on MNIST, highlighting its stability and minimal need for hyperparameter adjustment. FT and L1-sparse were trained for 10 epochs with learning rates in the range $[1 \times 10^{-2}, 1 \times 10^{-4}]$, with L1-sparse using a regularisation coefficient $\alpha$ in $[1 \times 10^{-4}, 1 \times 10^{-5}]$. GA and NegGrad+ were trained for 8 epochs, with learning rates in $[5 \times 10^{-4}, 1 \times 10^{-4}]$ also, NegGrad+ additionally used a loss weighting factor $\alpha$ in $[0.8, 0.9]$. IU required tuning $\alpha$ in the WoodFisher Hessian inverse approximation from 1 to 10. Random-label was trained for 10 epochs with learning rates in $[1 \times 10^{-3}, 1 \times 10^{-4}]$. SCRUB was trained for 10 epochs, with $\beta$ and $\gamma$ (KL divergence and classification loss weights) in $[0.3, 0.7]$ and learning rates in $[1 \times 10^{-3}, 1 \times 10^{-4}]$. For SalUn, we used 10 epochs, learning rates of $[1 \times 10^{-3}, 1 \times 10^{-4}]$, and sparsity ratios between $0.6$ and $0.7$. In Noisy-Scrubber, the noise module consists of two linear layers within each of the selector (64-dimensional), forget (512-dimensional), and retain (512-dimensional) neural networks. Training is performed for 10 epochs using a learning rate of $5 \times 10^{-4}$.

| Methods | Metrics | CIFAR10 | CIFAR100 | AGNews |
|---------|---------|---------|----------|--------|
| Retrain | RTE (sec) | 102.45 | 161.66 | 2466.45 |
| FT | ToW | 0.857±0.00 | 0.548±0.00 | 0.919±0.00 |
| | MIA-Gap | 0.123±0.00 | 0.317±0.00 | 0.028±0.00 |
| | RTE (sec) | 8.738±0.21 | 47.789±3.50 | 196.669±1.09 |
| GA | ToW | 0.035±0.00 | 0.669±0.00 | 0.027±0.00 |
| | MIA-Gap | 0.174±0.00 | 0.110±0.00 | 0.141±0.00 |
| | RTE (sec) | 6.195±0.12 | 43.647±4.01 | 150.834±0.40 |
| $\ell_1$-Sparse | ToW | 0.874±0.00 | 0.547±0.00 | 0.921±0.00 |
| | MIA-Gap | 0.109±0.00 | 0.316±0.00 | 0.021±0.00 |
| | RTE (sec) | 9.103±0.11 | 52.355±2.36 | 198.373±0.59 |
| IU | ToW | 0.074±0.00 | 0.047±0.0 | 0.626±0.00 |
| | MIA-Gap | 0.033±0.00 | 0.047±0.00 | 0.692±0.00 |
| | RTE (sec) | 8.898±0.76 | 48.584±1.92 | 78.316±2.35 |
| NegGrad+ | ToW | 0.777±0.00 | 0.602±0.00 | 0.917±0.00 |
| | MIA-Gap | 0.145±0.00 | 0.285±0.00 | 0.030±0.00 |
| | RTE (sec) | 9.660±0.58 | 27.322±1.22 | 141.388±3.32 |
| Random-Label | ToW | 0.353±0.00 | 0.640±0.00 | 0.027±0.00 |
| | MIA-Gap | 0.422±0.07 | 0.242±0.00 | 0.226±0.01 |
| | RTE (sec) | 10.343±0.08 | 53.796±3.89 | 249.117±0.54 |
| SCRUB | ToW | 0.794±0.00 | 0.589±0.0 | 0.406±0.10 |
| | MIA-Gap | 0.066±0.00 | 0.284±0.0 | 0.129±0.01 |
| | RTE (sec) | 9.251±0.64 | 52.307±2.51 | 238.013±2.96 |
| SALUN | ToW | 0.377±0.00 | 0.635±0.00 | 0.027±0.00 |
| | MIA-Gap | 0.565±0.00 | 0.244±0.00 | 0.557±0.01 |
| | RTE (sec) | 14.354±0.34 | 80.628±76.90 | 335.899±0.07 |
| Noisy-Scrubber | ToW | 0.907±0.00 | 0.687±0.00 | 0.969±0.00 |
| | MIA-Gap | 0.014±0.00 | 0.169±0.00 | 0.074±0.00 |
| | RTE (sec) | 17.521±1.78 | 25.511±0.23 | 97.126±1.25 |

Table 3: Performance comparison of MU methods in the random forgetting scenario. Results are reported as $a \pm b$, denoting the mean $a$ and standard deviation $b$ over 5 independent trials.

### A.6.4 MEMBERSHIP INFERENCE ATTACK (MIA) IMPLEMENTATION DETAILS

We used prediction confidence scores for membership inference attacks (MIA). An SVM-based attack model is trained using equal partitions from the retain and test sets, where the SVM receives the model's prediction confidences as input and predicts whether a sample was present in the training data. The attack model is defined as follows:

```
attack_model = SVC(C=3, gamma="auto", kernel="rbf")
```

After training the attack model, we use it to assess the membership of all samples in the forget set. Effective unlearning is indicated when these forget samples are classified as non-training data by the attack model. We measure performance as the ratio of true negatives (the number of forget samples correctly identified as non-training) to the total number of forget samples.

$$\text{MIA Performance} = \frac{TN_F}{|F|} \tag{9}$$

### A.6.5 ENVIRONMENT DETAILS

All experiments were conducted on a machine with the following configuration: NVIDIA RTX 4070 GPU (12GB), Intel i7-13700 13th Gen processor (5.1 GHZ), 32 GB of RAM, and running Ubuntu 24.04.2 LTS. We set up a Python virtual environment using conda 24.11.3 configured with Python 3.11.11, PyTorch 2.6.0, scikit-learn 1.6.1, and CUDA 11.8.

### A.7 RANDOM FORGETTING PERFORMANCE

Table 3 provides a comprehensive evaluation for the random forgetting scenario. On CIFAR10, CIFAR100, and AGNews, *Noisy Scrubber* achieves ToW gaps of 0.093, 0.313, and 0.031 versus retraining, demonstrating its ability to closely approximate exact unlearning. Furthermore, Noisy

| Methods | Metrics | MNIST | DbPedia | CIFAR100 (SWIN) |
|---|---|---|---|---|
| Retrain | **RTE (sec)** | 251.011 | 18104.17 | 758.76 |
| FT | **ToW** | 0.944±0.00 | 0.404±0.16 | 0.324±0.00 |
| | **MIA-Gap** | 0.00±0.00 | 0.527±0.14 | 0.417±0.00 |
| | **RTE (sec)** | 4.176±0.00 | 202.52±0.52 | 120.97±11.30 |
| GA | **ToW** | 0.0797±0.00 | 0.012±0.00 | 0.096±0.00 |
| | **MIA-Gap** | 0.114±0.00 | 0.00±0.00 | 0.00±0.00 |
| | **RTE (sec)** | 2.56±0.29 | 149.713±2.60 | 89.47±2.48 |
| $\ell_1$-Sparse | **ToW** | 0.920±0.00 | 0.714±0.02 | 0.321±0.00 |
| | **MIA-Gap** | 0.00±0.00 | 0.049±0.00 | 0.421±0.00 |
| | **RTE (sec)** | 5.683±1.23 | 201.948±4.71 | 119.15±2.69 |
| IU | **ToW** | 0.892±0.00 | 0.036±0.0 | 0.125±0.00 |
| | **MIA-Gap** | 0.038±0.00 | 0.00±0.00 | 0.00±0.00 |
| | **RTE (sec)** | 4.191±0.54 | 74.14±0.07 | 148.78±14.99 |
| NegGrad+ | **ToW** | 0.971±0.00 | 0.993±0.00 | 0.978±0.00 |
| | **MIA-Gap** | 0.003±0.00 | 0.00±0.00 | 0.00±0.00 |
| | **RTE (sec)** | 5.610±0.06 | 210.331±0.62 | 141.388±3.32 |
| Random-Label | **ToW** | 0.987±0.00 | 0.983±0.00 | 0.881±0.00 |
| | **MIA-Gap** | 0.00±0.00 | 0.00±0.00 | 0.00±0.00 |
| | **RTE (sec)** | 5.93±0.06 | 224.467±0.84 | 128.410±17.95 |
| SCRUB | **ToW** | 0.991±0.00 | 0.992±0.0 | 0.927±0.00 |
| | **MIA-Gap** | 0.00±0.00 | 0.00±0.0 | 0.00±0.00 |
| | **RTE (sec)** | 4.920±0.21 | 219.308±1.27 | 134.198±24.57 |
| SALUN | **ToW** | 0.988±0.00 | 0.993±0.00 | 0.879±0.00 |
| | **MIA-Gap** | 0.00±0.00 | 0.00±0.00 | 0.00±0.00 |
| | **RTE (sec)** | 6.976±0.03 | 261.42±35.51 | 172.006±158.80 |
| Noisy-Scrubber | **ToW** | 0.986±0.00 | 0.998±0.00 | 0.929±0.00 |
| | **MIA-Gap** | 0.00±0.00 | 0.00±0.00 | 0.00±0.00 |
| | **RTE (sec)** | 22.54±0.09 | 80.86±0.17 | 46.108±3.61 |

Table 4: Performance comparison of MU methods in the class-wise forgetting scenario. Results are reported as $a \pm b$, denoting the mean $a$ and standard deviation $b$ over 5 independent trials.

Table 5: Accuracy of Noisy Scrubber in the class-wise forgetting scenario. Results are reported as $a \pm b$, with $a$ representing the mean and $b$ the standard deviation computed over 5 independent trials. Full set represent full test dataset, forget and retain set represent forget and retain dataset extracted from test dataset.

| | | MNIST | CIFAR10 | CIFAR100 | CIFAR100 | AGNews | DBPedia |
|---|---|---|---|---|---|---|---|
| | | CNN | CNN | ResNet | SWIN | BERT | BERT |
| **Full Test Set** | Before | 99.06 | 79.45 | 59.91 | 61.15 | 94.41 | 99.24 |
| | After | 88.87±0.02 | 74.61±0.062 | 59.51±0.115 | 60.73±0.064 | 70.0±0.060 | 92.25±0.008 |
| | Retrained | 88.96 | 72.26 | 59.39 | 57.45 | 70.01 | 92.25 |
| **Forget Test Set** | Before | 99.93 | 57.30 | 17.67 | 49.0 | 98.95 | 97.98 |
| | After | 1.07±0.210 | 0.0±0.0 | 0.0±0.0 | 0.34±0.471 | 0.47±0.239 | 0.0±0.0 |
| | Retrained | 0.0 | 0.0 | 0.0 | 0.0 | 0.0 | 0.0 |
| **Retain Test Set** | Before | 99.07 | 81.91 | 60.34 | 61.27 | 92.89 | 99.34 |
| | After | 98.95±0.019 | 82.85±0.079 | 59.52±0.115 | 61.34±0.067 | 93.18±0.080 | 99.35±0.008 |
| | Retrained | 99.15 | 82.51 | 59.99 | 58.03 | 93.35 | 99.35 |

Scrubber consistently outperforms all baselines and prior state-of-the-art methods in ToW, attaining the highest scores on CIFAR10 (0.907), CIFAR100 (0.687), and AGNews (0.969). Privacy leakage, assessed by MIA-Gap, remains minimal (maximum 0.169) across all datasets, suggesting strong resistance against membership inference attacks. Among approximate methods, Noisy Scrubber achieves superior forget quality, utility, and generalisation. In terms of computational efficiency, Noisy Scrubber delivers significant runtime improvements over retraining, particularly on large models like ResNet (CIFAR100) and BERT (AGNews).

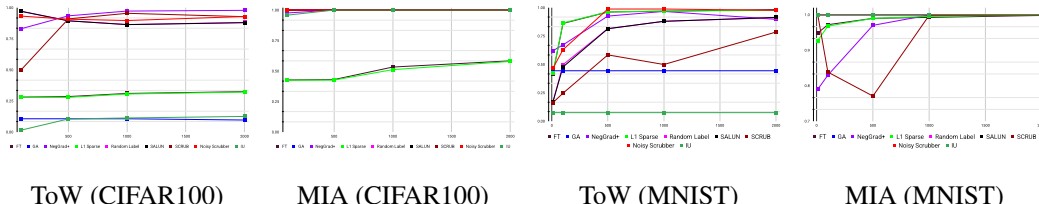

| ToW (CIFAR100) | MIA (CIFAR100) | ToW (MNIST) | MIA (MNIST) |

Figure 5: Comparison of the impact of number of retain samples on ToW and MIA performance for CIFAR100 (SWIN) and MNIST datasets

### A.8 CLASS-WISE FORGETTING PERFORMANCE

Table 4 presents a comprehensive evaluation on the CIFAR100 (SWIN), MNIST, and DBPedia datasets. The results demonstrate that Noisy Scrubber enables efficient unlearning, supporting the findings discussed in the experimental results section of the paper. Table 5 summarises the results on the forget and retain subsets derived from the test set, which indicates that the method closely matches the retrained model's behaviour across the forget, retain, and test sets, demonstrating effective unlearning.

### A.9 COMPARISON: NUMBER OF RETAIN SAMPLES

Figure 5 shows that on CIFAR100, increasing the number of retain samples initially improves ToW performance, followed by a decline when the retain set reaches 1,000 samples. This drop occurs because the forget set consists of only 500 samples from a single class. As the retain set grows, the noise module becomes biased, raising accuracy on the forget set and reducing ToW. With further increases in retain samples, Noisy Scrubber's performance recovers. For MNIST Figure 5, Noisy-Scrubber starts with lower ToW compared to other methods when trained with very few retain samples, indicating that the noise module initially requires few data samples for effective learning. Additionally, both figures indicate that Noisy Scrubber consistently maintains MIA scores close to 1 across datasets, confirming its robustness to membership inference attacks and its ability to prevent information leakage.

### A.10 USE OF LARGE LANGUAGE MODELS(LLMS)

We used LLMs solely for non-technical assistance in preparing this paper. Specifically, LLMs were used for polishing grammar and improving readability of text, identifying related works during the literature survey and summarizing them. No LLMs were used for generating novel research ideas, designing experiments. All scientific contributions are original to the authors.

