# OpenReview forum: "Noisy Scrubber: Unlearning Using Noisy Representations"
_ICLR.cc/2026/Conference — Submitted to ICLR 2026_

### Official Review · Reviewer_sSWw · 2025-10-28

**Soundness:** 2
**Presentation:** 3
**Contribution:** 2
**Rating:** 2
**Confidence:** 4

**Summary:**

Proposes a Noisy Scrubber — a lightweight, pluggable module that injects noise into intermediate representations. The module $ p(\cdot, \phi)$ is trained for producing the noise that is then added to intermediate representations, so that the model’s output becomes uncertain on forgetting samples while remaining unchanged on retaining data. A distilled version (Distill-NS) merges the effect of both $\theta$ and $\phi$ into a single unified model via knowledge distillation.

**Strengths:**

1. Clear writing. Shifting unlearning focus from parameter manipulation to representation-space noise injection.

2. The proposed method is pluggable, data-efficient, and architecture-agnostic.

3. Experiments cover vision and NLP datasets, including both qualitative and quantitative metrics.

**Weaknesses:**

1. Around line 222, "In practice, strong convexity can be introduced during training through the addition of L2 regularisation." Does strong convexity refer to the global convexity of a deep neural network's loss function? Is there a reference that proves this statement?

2. The theory objective seems mismatched with the unlearning training objective. The bounds assume that the outputs approximate retraining, but the training uses the UEO loss. There is no justification that retraining produces UEO-like outputs.

3. Retrained models typically distribute probabilities unevenly across similar classes. Forcing uniformity could eliminate meaningful structure and hinder generalization. This also leads to a mismatch between the theory and the training objective.

4. This work introduces a learnable plug-and-play network for learning an extra objective other than the original model's, an idea that is quite similar to LoRA. However, the paper lacks a discussion and comparison with similar work based on LoRA [1].

[1] Cha S, Cho S, Hwang D, et al. Towards robust and parameter-efficient knowledge unlearning for LLMs. ICLR 2025

**Questions:**

Please refer to the weaknesses.

---

> ### Author Response · Authors · 2025-11-28
>
> Thanks to Reviewer sSWw, we appreciate your efforts in reviewing our work.
>
> # Responses to weaknesses
>
> **W1:** No, strong convexity does not refer to global strong convexity it is referring to piecewise or local strong convexity [1]. Addition of L2 regularization during training, helps the loss function exhibit piecewise strong convexity along the optimization path, aiding stability [1]. Rather than stating "strong convexity" we shall clarify that it makes the loss landscape piecewise strongly convex, as discussed in [1].
>
> **W2:** In theory we showed that we can find a perturbation that makes output of noisy scrubber closer to retrained model ie. find $\epsilon$ that minimizes $||g(h(x, \theta^t_{\text{feat}}) + \epsilon, \theta^t_{\text{cls}}) -f(x,\theta^{r})||$. To solve for above objective we need retrained model $\theta^r$ which is not available so we model retrained model outputs using UEO distribution.
>
> In class-wise forgetting, the model is retrained with an entire class removed, so any input from that class whether it was in the original training forget set or is a held-out test sample, sees the same lack of training signal, therefore gives low confidence (high uncertainty) for both sets, making them indistinguishable. In random-sample forgetting, retraining typically reduces the model’s probability on the true label and spreads probability mass more evenly over the other classes (an entropy-increasing or near-uniform output), which also lowers confidence on forgotten examples and makes their prediction statistics very similar to unseen test examples.
>
> The UEO distribution ensures that the model's prediction is moved away from the true class label and distributed more equally across other classes. This effectively lowers the model's confidence in the true class prediction for the forgotten sample.
>
> **W3:** We acknowledge that a retrained model might exhibit non-uniform probabilities due to shared visual patterns correlations (example dog and wolf share visual patterns which will result in non-uniform probability if we forget one of class). Explicitly modeling this unknown structure without access to the retrained model is intractable (predicting exactly which class model will confuse). We employ the UEO distribution assuming that a model which has truly "forgotten" a class should exhibit maximal uncertainty regarding it. UEO ensures aggressive forgetting while retaining the utility required for correct classification among the remaining classes. Our results confirm high utility and generalization, as reflected by the consistently high ToW scores across all datasets.
>
> **W4:** We acknowledge that the lightweight, attachable noise module in Noisy Scrubber has structural similarities to parameter-efficient fine-tuning techniques such as LoRA. However, our work differs fundamentally in scope and intent. Noisy Scrubber is designed as a general-purpose machine unlearning framework applicable across diverse modalities and architectures, including vision models (e.g., ResNet-101 and SWIN Transformer) and text models (e.g., BERT). In contrast, the LoRA-based unlearning approach by Cha et al. (2025)[2] targets knowledge unlearning tailored for large language models only. Our primary contribution is establishing a scalable, representation-level perturbation method for general machine unlearning tasks, with the adaptation of Noisy Scrubber to large-scale foundation models and LLMs explicitly noted as future work.
>
> # Responses to questions
>
> please refer responses to weaknesses above
>
> # References
>
> [1] Milne, Tristan. "Piecewise strong convexity of neural networks." Advances in neural information processing systems 32 (2019)
>
> [2] Cha S, Cho S, Hwang D, et al. Towards robust and parameter-efficient knowledge unlearning for LLMs. ICLR 2025

---

### Official Review · Reviewer_oLaF · 2025-10-31

**Soundness:** 2
**Presentation:** 3
**Contribution:** 1
**Rating:** 2
**Confidence:** 4

**Summary:**

The paper proposes Noisy Scrubber, which is a plug-in module that injects targeted noise into latent representations to unlearn a forget set F. The proposed model seeks to present an unlearning method that avoids changing the parameters of the model, but instead, for the samples in F, it adds noise to the latent representation to maximize the entropy of the prediction when the last layer is applied to them.  The paper provides some theoretical bounds under strong convexity and Lipschitz/smoothness assumptions. Experiments on CIFAR-10/100, AGNews, DBPedia with CNN/ResNet/SWIN/BERT report strong utility and privacy.

**Strengths:**

- The idea is straightforward and architecture-agnostic.
- The method is easy to implement and enough details have been added by the authors.
- The paper is clearly written and easy to follow.

**Weaknesses:**

1. Although the theoretical results are claimed as the main contributions,  Proposition 1 is not new and has been presented by prior work on certified unlearning. For example lemma 8 in [1] derives the exact bound that you have; the only difference is that the bound you have is scaled by |F| because the bound in [1] is for the case in which the number of samples in two datasets differ by 1, but in your case they differ by |F| (the number of forget samples). The authors fail to reference prior works on machine unlearning that introduce the same bounds and instead introduce them as if they are new contributions. After the work in [1] there are newer works on certified machine unlearning that relax the assumptions about strong convexity, which do not apply to the models used in practice [2].
2. Authors in line 222 mention that “in practice strong convexity can be introduced during training through the addition of L2 regularization.” Addition of L2 regularization makes the objective strongly convex only when the objective is convex. That is not the case for deep neural networks that are non-convex. Strong convexity in general is a very restrictive assumption, which is why in works such as [2] on certified unlearning, the focus is on relaxing this condition.
3. The justification for avoiding a change in the model parameters for unlearning is not clear. In the worst-case scenario that an adversary gains access to the model they can easily recover information about the forget samples. As mentioned by the authors, the goal in machine unlearning is to derive model parameters that are indistinguishable from the parameters of a retrained model. If the goal is to just perform post-hoc unlearning with the maximum uncertainty principle used by authors, is there any advantage over the following pseudocode:
- Compute $y = f(x,\theta^t)$
- If $||x-x'|| < \epsilon$ for some $x' \in F$:
    - Return a label other than $y$ chosen uniformly at random
- Else:
    - Return $y$

I claim this is what your method is ideally supposed to achieve. It tries to train a second neural network such that it generates near-0 noise for the samples not in $F$ to get the same prediction and return add noise to $x \in F$ such that the prediction becomes like a random predictor with uniform probability over the remaining classes. With the above pseudo-code we do not make any additional assumptions compared to what you make, and achieve the same goal.

4. The SVM-based MIA that is used for evaluations is very weak compared to the SOTA MIA methods in the literature. I encourage the authors to utilize SOTA MIA for their evaluations rather than relying on basic approaches. [4] introduces is one of the SOTA MIAs that is an adaptation of  [3], but more practical with a few shadow models. There are also MIAs designed specifically to evaluate unlearning methods [5,6].
5. There are more recent works on machine unlearning for classification models that have not been used as base-lines in the experiments. Please see [2,7,8,9,10].
6. The bound in proposition 2, which again relies on model smoothness and strong convexity assumptions, is exponential in the number of gradient descent steps (which is large in practice). This makes the bound vacuous and not useful in practice. The provided bound also doesn’t depend on the value of $\epsilon$ at all. It basically is a replica of proposition 1 because authors just assume $\epsilon=0$, which basically again gives us the difference of the original model and retrained model in equation $1$. So i think in general this proposition does not provide us with any useful and meaningful information about the unlearned model
7. In general recent works on unlearning [7] has shown that  the assumption in line 233 about the retrained model regarding their uncertainty on the forget samples is not accurate. In forgetting a random set of samples a retrained model would behave similarly on the forget samples as the test samples, which is why MIAs would achieve an AUC of 50% on detecting forget samples vs. test samples.


[1] Neel, S., Roth, A., & Sharifi-Malvajerdi, S. (2021, March). Descent-to-delete: Gradient-based methods for machine unlearning. In Algorithmic Learning Theory (pp. 931-962). PMLR.

[2] Zhang, B., Dong, Y., Wang, T., & Li, J. (2024). Towards certified unlearning for deep neural networks. arXiv preprint arXiv:2408.00920.

[3] N. Carlini, S. Chien, M. Nasr, S. Song, A. Terzis and F. Tramèr, "Membership Inference Attacks From First Principles," 2022 IEEE Symposium on Security and Privacy (SP), San Francisco, CA, USA, 2022, pp. 1897-1914, doi: 10.1109/SP46214.2022.9833649.

[4] Zarifzadeh, S., Liu, P., & Shokri, R. (2024, July). Low-cost high-power membership inference attacks. In Proceedings of the 41st International Conference on Machine Learning (pp. 58244-58282).

[5] Hayes, J., Shumailov, I., Triantafillou, E., Khalifa, A., & Papernot, N. (2025, April). Inexact unlearning needs more careful evaluations to avoid a false sense of privacy. In 2025 IEEE Conference on Secure and Trustworthy Machine Learning (SaTML) (pp. 497-519). IEEE.

[6] Cadet, X. F., Borovykh, A., Malekzadeh, M., Ahmadi-Abhari, S., & Haddadi, H. (2025, June). Deep Unlearn: Benchmarking Machine Unlearning for Image Classification. In 2025 IEEE 10th European Symposium on Security and Privacy (EuroS&P) (pp. 939-962). IEEE.

[7] Ebrahimpour-Boroojeny, A., Sundaram, H., & Chandrasekaran, V. Not All Wrong is Bad: Using Adversarial Examples for Unlearning. In Forty-second International Conference on Machine Learning.

[8] Cha, S., Cho, S., Hwang, D., Lee, H., Moon, T., & Lee, M. (2024, March). Learning to unlearn: Instance-wise unlearning for pre-trained classifiers. In Proceedings of the AAAI conference on artificial intelligence (Vol. 38, No. 10, pp. 11186-11194).

[9] Bonato, J., Cotogni, M., & Sabetta, L. (2024, September). Is retain set all you need in machine unlearning? restoring performance of unlearned models with out-of-distribution images. In European Conference on Computer Vision (pp. 1-19). Cham: Springer Nature Switzerland.


### Minor weaknesses:

1. I recommend authors to use bold font to show the best result in each column for their presented table and maybe an underlined one for the second-runner. It is difficult to read the tables.
2. line 284 (where a specified …) seems incomplete. I think some words are missing.

**Questions:**

1. What is the advantage of using a plug-in module that disrupts the correct prediction in the last layer by adding noise to the representation layer over a post-hoc module that checks for the vicinity to the samples in $F$ and then makes a wrong prediction if it is nearby to sample in $F$? Why the need for a separate neural network that is supposed to perform this task (see 3 in weaknesses for more details)? Could the authors perform some analysis to show how the nearby region of the forget samples transform under this modification to the representation space? That could provide some insights into its usefulness.

2. What is the use-case of proposition 2 that does not depend on $\epsilon$ and is exponential in the number of gradient decent steps?

3. Please also see the weaknesses and see if they can be addresses.

---

> ### Author Response · Authors · 2025-11-28
>
> Thanks to Reviewer oLaF.
>
> # Response to weaknesses
>
> **W1:** Our primary contribution of Proposition 1 lies in first part, derived without assuming convexity. We showed, under strong convexity, bound becomes tighter but we did not use it in any analysis later. We will cite [1] and acknowledge that under strong convexity, second part of our Proposition 1 coincides with Lemma 8 in [1]. Regarding [4], their Theorem 3.4 relies on boundedness assumption $||w^* - \tilde{w}^*|| \leq 2C$ (Eq.26 in Appendix A.4). In contrast, we derive explicit expansion of this measure using SGD dynamics.
>
> **W2:** Addition of L2 regularization during training help loss function exhibit piecewise strong convexity along optimization path, aiding stability [2]. We will clarify that it makes loss landscape piecewise strongly convex and refer [2].
>
> **W3:** We consider online machine unlearning setting, where forget requests arrive continuously and must be handled as they come. Our method addresses this by training small noise module providing immediate, temporary unlearning. We also acknowledge adversary with access to model may recover information about the forgotten samples. To mitigate, Distill-NS can be used periodically. Rigorous threat model and robustness evaluation can be part of future work.
>
> Advantage over given algorithm:
>
> - Given algorithm is inference only algorithm, which require forget set $F$ during inference and defeats purpose of unlearning.
> - Algorithm is dependent on $x^\prime$ leading to reduce in privacy utility where samples although not close to chosen $x^\prime$ is still part of forget set $F$.
>
> **W4:** SVM-based confidence attack is standard benchmark used in many prior unlearning works. We utilized SVM-based MIA adapted from SalUn [3]. We agree that incorporating more advanced MIAs would strengthen the evaluation and will pursue in future work.
>
> **W5:** We evaluated Noisy Scrubber against 8 diverse methods to cover primary unlearning strategies, including fine-tuning, knowledge distillation, gradient manipulation, and sparse parameter updates. [4] focuses on certified unlearning that prioritizes theoretical guarantees over scalability. [5] and [6] rely on direct modification of model parameters, category we already extensively compared. We agree that incorporating more comparisons would strengthen the evaluation and will pursue in future work.
>
> **W6:** The proposition should be read as a two-sided, constructive statement. It gives provable upper bound on distance between noise-perturbed output and retrained model output, while trivial lower bound $z(\epsilon) = ||g(h(x, \theta^t_{\text{feat}}) + \epsilon, \theta^t_{\text{cls}}) -f(x,\theta^{r})|| \geq 0$ completes the range of possible values. Upper bound is a worst-case certificate that holds for any $\epsilon$. This establishes an attainable interval for output difference and motivates optimizing $\epsilon$ to make objective small. Under additional assumption that $g$ is a linear layer, $z(\epsilon)$ is convex in $\epsilon$. Standard gradient-based methods will thus find $\epsilon$, which pushes perturbed model’s outputs arbitrarily close to retrained model.
>
> **W7:** We acknowledge retrained model would behave similarly on forget samples as test samples. We will revise the section to provide a clearer explanation of high-uncertainty assumption for forgotten samples. In class-wise forgetting, model is retrained with an entire class removed, so any input from that class whether it was in forget set or test set, sees same lack of training signal, giving low confidence (high uncertainty) for both sets, making them indistinguishable. In random-sample forgetting, retraining typically reduces model’s probability on true label and spreads probability mass over other classes. The UEO distribution ensures that model's prediction moved away from true class label and distributed equally across other classes. This lowers the model's confidence in true class for the forgotten sample and makes prediction statistics similar to unseen test examples.
>
> # Responses to questions
>
> **Q1:** Please refer W3
>
> **Q2:** Please refer W6
>
> **Q3:** Please refer to weaknesses.
>
> # References
>
> [1] Neel, S., Roth, A., & Sharifi-Malvajerdi, S. (2021, March). Descent-to-delete: Gradient-based methods for machine unlearning. In Algorithmic Learning Theory. PMLR.
>
> [2] Milne, Tristan. Piecewise strong convexity of neural networks. NeurIPS (2019)
>
> [3] Fan, Chongyu, et al. Salun: Empowering machine unlearning via gradient-based weight saliency in both image classification and generation. ICLR (2024)
>
> [4] Zhang, B., Dong, Y., Wang, T., & Li, J. (2024). Towards certified unlearning for deep neural networks.
>
> [5] Ebrahimpour-Boroojeny, A., Sundaram, H., & Chandrasekaran, V. Not All Wrong is Bad: Using Adversarial Examples for Unlearning. 42nd ICML.
>
> [6] Cha, S., Cho, S., Hwang, D., Lee, H., Moon, T., & Lee, M. (2024, March). Learning to unlearn: Instance-wise unlearning for pre-trained classifiers. AAAI.

---

### Official Review · Reviewer_8xtF · 2025-11-02

**Soundness:** 3
**Presentation:** 2
**Contribution:** 2
**Rating:** 4
**Confidence:** 4

**Summary:**

The paper proposes Noisy Scrubber, an approximate machine-unlearning (MU) method that doesn’t modify model weights. Instead, it adds a small learned “noise module” that perturbs intermediate representations so the model behaves like it had been retrained without the forget set. The paper also provides two bounds. 1. A parameter-gap bound between trained vs. retrained models and 2. An output-gap bound.

**Strengths:**

1.The paper proposed a formulation of approximate unlearning via an attachable, learnable noise module without retraining the model.
2.Providing the clear theoretical analysis that connects the data removal size to output and parameter deviation, providing interpretability uncommon in empirical MU works.
3.Conducting comprehensive experiments across several datasets with strong ToW and MIA metrics.

**Weaknesses:**

1.The theoretical analysis relies on strong convexity, which cannot directly be applied to deep neural networks, limiting the analysis’s value.
2.In proposition 1, the author proves the parameter-gap bound with several assumptions, like L-smooth, T steps, and step size /alpha, but does not talk about the correlation between those parameters. For example, the author can use Gronwall inequality here.
3.Regarding the UEO objective, the optimization primarily constrains the output probabilities in the simplex space, pushing the predicted distribution toward uniformity except for one entry.
4.In proposition 2, the output-gap bound assumes the existence of an optimal perturbation \epsilon such that the noisy model output approximates the retrained model’s output. This implicitly presumes a controlled Lipschitz behavior of the composed mapping g(h(x,\theta)), yet no quantitative bound on its Jacobian norm or layer-wise Lipschitz constants is given.

**Questions:**

1.Could the author explain how those parameters interact in proposition 1?
2.In proposition 2, the output-gap bound relies on the Lipschitz continuity of g(h(x,\theta)); has the author considered bounding the Jacobian norms to make the result quantitatively evaluable?

---

> ### Author Response · Authors · 2025-11-28
>
> Thanks to Reviewer 8xtF, we appreciate your efforts in reviewing our work.
>
> # Responses to weaknesses
>
> **W1:** We acknowledge that deep neural networks are generally non-convex. However, Proposition 1 provides two bounds:
>
> - The first, more general bound, relies only on the M-Lipschitz smooth assumption.
>   $$||\theta^t - \theta^r|| \leq \frac{2G|F|}{M|D|} (e^{\alpha MT} - 1) $$
> - The second, tighter bound, relies on $\mu$-strong convexity.
>   $$||\theta^t - \theta^r|| \leq \frac{G|F|}{\mu|D|} $$
>
> We just showed that under strong convexity the bound becomes tighter but we did not use it in any analysis later.
>
> **W2:** Parameters $\alpha$, $M$ and $T$ are independent and no correlation between them is assumed. Although we can consider some $\alpha$ that correlates with $M$ or $T$ or both which can tightens the bound further. For example we can choose $\alpha = \frac{1}{M}$ or $\alpha = \frac{1}{MT}$ which further tightens bound.
>
> **W3:** In class-wise forgetting, the model is retrained with an entire class removed, so any input from that class whether it was in the original training forget set or is a held-out test sample, sees the same lack of training signal, therefore gives low confidence (high uncertainty) for both sets, making them indistinguishable. In random-sample forgetting, retraining typically reduces the model’s probability on the true label and spreads probability mass more evenly over the other classes (an entropy-increasing or near-uniform output), which also lowers confidence on forgotten examples and makes their prediction statistics very similar to unseen test examples.
>
> The UEO distribution ensures that the model's prediction is moved away from the true class label and distributed more equally across other classes. This effectively lowers the model's confidence in the true class prediction for the forgotten sample.
>
> **W4:** The existence of a Lipschitz constant (which is standard in convergence analyses) mathematically implies a bound on the Jacobian norm $||J_f(x)||\leq L$, introducing a separate quantitative bound on the Jacobian would be redundant to the Lipschitz assumptions.
>
> In our experiments the classification head $g(⋅)$ is a linear layer. Since we employ weight decay during training, the spectral norm $||W||_2$ and thus the Jacobian is implicitly bounded and well-controlled in practice.
>
> # Responses to questions
>
> **Q1:** Parameters $\alpha$, $M$ and $T$ are independent and have no correlation between them is assumed. [Please refer W2 for more details]
>
> **Q2:** In our experiments the classification head $g(⋅)$ is a linear layer. Since we employ weight decay during training, the spectral norm $||W||_2$ and thus the Jacobian is implicitly bounded and well-controlled in practice.

---

### Official Review · Reviewer_QViu · 2025-11-04

**Soundness:** 3
**Presentation:** 3
**Contribution:** 3
**Rating:** 6
**Confidence:** 2

**Summary:**

The paper presents an alternative strategy to unlearning where perturbation is used to change the latent representations using an optimization of knowledge distillation loss function with respect to the exact unlearned model.

**Strengths:**

+ good use of loss-based noisy generation
+ interesting design of the noise perturbation and model - independent parameters

**Weaknesses:**

- use of rather old models and datasets, with less diversity
- Parallels with differential privacy and MIA resistance not explored
- Limited evaluations against various unlearning methods

**Questions:**

The paper’s way for exploring unlearning using model-independent and distance-based parameter perturbation is interesting and I enjoyed reading on the approach. I saw some parallels with using DP to avoid MIA, though not much investigated in the paper or mentioned.

I think the noise module being independent of the model is an interesting and useful feature, though I was not sure how it  “allows us to forget learned knowledge for a specific period of time”.

---

> ### Author Response · Authors · 2025-11-28
>
> Thanks to Reviewer Qviu, we appreciate your efforts in reviewing our work.
>
> # Responses to weaknesses
>
> **W1:** We respectfully disagree that our evaluation lacks diversity. Our experiments spanned both vision (MNIST, CIFAR-10, CIFAR-100) and text (AGNews, DBPedia) modalities. Crucially, we tested a range of architectures, from simple CNNs to large and complex models including ResNet-101, SWIN Transformer, and BERT. This robust evaluation across large architectures in both image and text domains highlights the scalability and wide applicability of Noisy Scrubber. We used datasets and models widely used for benchmarking unlearning in classification tasks some examples are [1, 2, 3].
>
> **W2:** We respectfully clarify that resistance to Membership Inference Attacks was a central component of our evaluation as demonstrated in Section 5.2 and Tables 1, 3, and 4 where we rigorously quantified privacy leakage using the MIA-Gap metric and an SVM-based confidence attack. These results confirm that Noisy Scrubber achieves negligible leakage that is comparable to exact retraining. Regarding Differential Privacy (DP), rather than satisfying the worst-case theoretical bounds of DP, our work targets the approximate unlearning paradigm where the objective is indistinguishability from retrained model behavior as theoretically bounded in Proposition 2 and trivial lower bound $||g(h(x, \theta^t_{\text{feat}}) + \epsilon, \theta^t_{\text{cls}}) -f(x,\theta^{r})|| \geq 0$. While DP provides strict guarantees, it often necessitates trade-off with model utility. We prioritized empirical verification via MIA to demonstrate that our method does not leak more information than the gold standard of full retraining.
>
> **W3:** We compared Noisy Scrubber against eight established approximate unlearning methods, including recent approaches like SalUN and SCRUB. These baselines (FT, GA, $\ell_1$-sparse, IU, NegGrad+, Random-Label, SCRUB, and SALUN) cover diverse strategies from fine-tuning to knowledge distillation and gradient manipulation.
>
> # Responses to questions
>
> **Q1:** Rather than satisfying the worst-case theoretical bounds of DP, our work targets the approximate unlearning paradigm where the objective is indistinguishability from retrained model behavior as theoretically bounded in Proposition 2 and trivial lower bound $||g(h(x, \theta^t_{\text{feat}}) + \epsilon, \theta^t_{\text{cls}}) -f(x,\theta^{r})|| \geq 0$. While DP provides strict guarantees, it often necessitates trade-off with model utility. We prioritized empirical verification via MIA to demonstrate that our method does not leak more information than the gold standard of full retraining. We will explore parallels with DP for deriving certified unlearning guarantee in future work.
>
> **Q2:** Noise module allows "reversible unlearning" or forgetting for a specific period of time. The key advantage is that the noise module $\phi$ is lightweight, attachable, and independently trained. The unlearned model parameters are defined as $\theta^u = [\theta^t, \phi]$. Since the core trained model parameters ($\theta^t$) are frozen and not directly modified during unlearning, the effect of unlearning is contingent upon the attached noise module. Reversible unlearning is achieved because the noise module can be easily detached or toggled off, instantly returning the system to the original trained state ($\theta^t$).
>
> # References
>
> [1] Chien, Eli, et al. "Certified machine unlearning via noisy stochastic gradient descent." Advances in Neural Information Processing Systems 37 (2024)
>
> [2] Zhao, Kairan, et al. "What makes unlearning hard and what to do about it." Advances in Neural Information Processing Systems 37 (2024)
>
> [3] Fan, Chongyu, et al. "Salun: Empowering machine unlearning via gradient-based weight saliency in both image classification and generation.

---

### Meta-Review · Area_Chair_2MXX · 2026-01-06

**Summary:**

I would be recommending a reject based on the reviewer concerns (which I agree with) regarding the use of the "uniform-except-one" or UEO unlearning objective. It makes sense to either try to match the parameters of the retrained model, or match the output of the retrained model. However, in both cases, it is not quite straightforward to do so in the absense of the retrained model, and the overall goal of approximate unlearning is to efficiently modify the model to behave as the retrained model without having access to the retrained model.

However, the use of UEO as the unlearning objective makes it quite unrelated to the objective considered in the theoretical analysis (proposition 2). It would be good to have a model that has the properties as promised by the analysis, but that objective is not tractable without retraining, making it somewhat vacuous. Alternately, there is no concrete analysis tying the predictions of the retrained model to the training target UEO objective (beyond some verbal motivation which is not appropriately justified in my opinion). This was also raised by Reviewer oLaF: _"[..] the assumption in line 233 about the retrained model regarding their uncertainty on the forget samples is not accurate."_

One can further argue that something like the UEO objective is actually counter to the unlearning goals, as an adversary can check the predictions of a sample and use the uniformity of the model predictions as a signal for whether that example is in the forget set or not. If the unlearning is completely successful, then forgetten examples would have UEO predictions, which would be easy to detect.

**Reviewer Concerns:**

Overall, to the best of my understanding, the main concerns were the following:

- **Use of UEO as the unlearning objective as a surrogate for the target objective of matching the retrained model's predictions.**
  - This issue was not appropriately addressed by the authors in my opinion, and I have provided further details in the above section. The use of this problematic objective is my main justification for recommending a rejection.
- **Weak evaluation on old models and datasets.**
  - In my opinion, this concern was not justified as the submission covers both text and vision learning problems, and a variety of architectures.
  - In fact, according to reviewer 8xtF, one of the strengths of this submission was _"Conducting comprehensive experiments across several datasets with strong ToW and MIA metrics."_
- **Propositions 1 and 2 rely on various assumptions not usually satisfied in practice with the models usually considered, and the novelty of the analyses was not clear given existing literature.**
  - This concern was only partially addressed by the authors by highlighting that the assumptions considered are somewhat standard in unlearning or optimisation analyses.
  - The novelty of proposition 1 was brought into question but the authors clarify that they consider a different set of assumptions (Lipschitz-smoothness and bounded gradients) as opposed to bounded parameter change (considered in existing literature such as Zhang et al. (2024)). However, it is important to note that the assumptions of Proposition 1 ($T$ step gradient descent and bounded gradients) imply the bounded parameter change assumptions, thus making the assumptions of Proposition 1 more restrictive than those of existing literature.
  - Furthermore, the way the results are presented, it is not clear how Proposition 1 is related to the proposed scheme or how Proposition 1 and Proposition 2 can be compared as they are bounding different things (even though the upper bounds look quite similar).
  - As noted by multiple reviewers, the results of Proposition 2 appear too loose and somewhat vacuous as the effect of $\epsilon$ (or the value of the mismatch objective at $\epsilon$) is not clear in the upper bound. There appears to be an assumption that the optimal objective value at the argmin is zero but that may not always be possible for any $x$ (unless we are assuming an extremely expressive model to create the perturbations). The authors provide a response, but the concern did not seem to be properly addressed in my opinion.
- **Various recent unlearning baselines and MIA schemes were not considered.**
  - The authors provided justification for the use of the SVM-based MIA even though reviewers cited papers highlighting the unreliability of such MIA schemes.
  - The authors also considered various unlearning schemes even though they did not consider all the ones raised by the reviewers.

**Reviewer Scores:**

Based on the responses, I think that reviewer QViu (original score 6) would raise their score slightly as their concerns were addressed the best. Reviewers 8xtF (score 4), oLaF (score 2) and sSWw (score 2) might have also raised their scores, but only slightly as their main concerns were not addressed appropriately in my opinions.

---

### Decision · Program_Chairs · 2026-01-26

Reject